# VELR: Efficient Video Reward Feedback via Ensemble Latent Reward Models

**Liyu Zhang** [1 2]   **Kehan Li** [2]   **Tao Zhou** [2]   **Zeyi Huang** [2]   **Chao Li** [1]   **Jiming Chen** [1]

## Abstract

Reward feedback learning (ReFL) is effective for both text-to-image (T2I) and text-to-video (T2V) generation with image reward models (RMs). However, image RMs are misaligned with temporal objectives of T2V, motivating ReFL with video reward models. Nsevertheless, directly deploying video RMs is impractical due to their large parameter size and the prohibitive cost of memory. To address this, we propose VELR: an efficient framework that employs ensemble latent reward models (LRMs) to predict rewards directly in latent space, bypassing expensive backpropagation through VAE decoders and video RMs. Specifically, we introduce the ensemble technique for the LRM, which enhances capacity, quantifies uncertainty, and mitigates reward hacking. VELR achieves a reduction of up to 150GB in memory, requiring as little as 12.4% of the memory compared to standard ReFL. Experiments on OpenSora-1.2, CogVideoX-1.5, and Wan-2.1 with large-scale video RMs demonstrate that VELR achieves comparable performance as standard ReFL and enables efficient and robust video RM-based ReFL at scales previously unattainable.

## 1. Introduction

Recent advances in text-to-video (T2V) generation have yielded increasingly powerful models (Hong et al., 2023; Zheng et al., 2024; Wan et al., 2025). Despite this progress, such models often exhibit misalignment with human preferences and may produce undesirable, toxic, or harmful contents. Reinforcement learning with human feedback (RLHF) has therefore emerged as a critical paradigm for aligning foundation models with human expectations, demonstrating notable success in both language (Christiano et al., 2017;

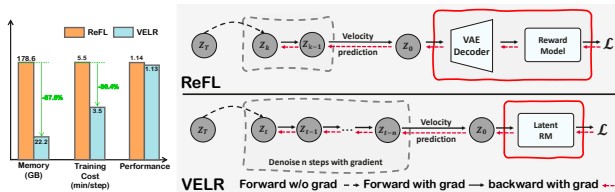

*Figure 1.* We propose **VELR**, a novel reward feedback learning framework that utilizes the ensemble latent reward model.

Ouyang et al., 2022) and visual domains (Lee et al., 2023).

In the context of T2V generation, several RLHF approaches have been explored, including policy optimization methods such as DDPO (Black et al., 2024), and GRPO (Xue et al., 2025; Liu et al., 2025a), as well as preference optimization methods such as DPO (Wallace et al., 2024; Wu et al., 2025c). While these methods can improve alignment, they typically incur substantial computational costs and often yield minor performance gains, limiting their scalability.

Reward Feedback Learning (ReFL) has recently been proposed as a more direct and effective alternative (Xu et al., 2023), where reward signals are backpropagated to the generative model to shape its preferences. However, existing ReFL methods require maintaining gradients through both the VAE decoder and the reward model, leading to excessive memory overhead. Consequently, existing ReFL solutions rely on image reward models (RMs) and require extensive memory optimizations (Yuan et al., 2024; Prabhudesai et al., 2025). Such image-based ReFL methods are insufficient for T2V models, as they fail to capture temporal consistency, which is critical for video generation.

For T2V models, this limitation naturally motivates the use of video-based reward models. and a recent work has demonstrated that fine-tuning with Vision-Language Models (VLMs) as video reward models can significantly improve video generation quality (Wu et al., 2025b). However, state-of-the-art video reward models (Liu et al., 2025b; Wang et al., 2025b) are typically built on large-scale VLM backbones (Bai et al., 2025), making them prohibitively expensive to train. ReFL training on a single video frame can require more than 100 GB of memory, making ReFL impractical on most hardware.

[1]College of Control Science and Engineering, Zhejiang University, Hangzhou, China. [2]Central Research Institute, Huawei, Hangzhou, China. Correspondence to: Chao Li <chaoli@zju.edu.cn>.

*Proceedings of the 43rd International Conference on Machine Learning*, Seoul, South Korea. PMLR 306, 2026. Copyright 2026 by the author(s).

A promising direction to address this challenge is to bypass both the decoder and reward model by training **latent reward models (LRMs)** that predict rewards directly from latent representations (Ding et al., 2025; Zhang et al., 2025). Ding (Ding et al., 2025) first introduced the concept of LRM. However, their implementation relies on a simple network architecture and is evaluated only with image-based reward models on consistency models. While these explorations highlight the potential of LRMs for efficient ReFL, existing approaches fall short of providing reliable solutions for large-scale video reward models. Specifically, the LRM in (Ding et al., 2025) is insufficient to estimate large-scale video RMs and exhibits poor generalization to out-of-distribution samples, leading to unstable reward signals.

To address these limitations, we propose VELR: Efficient **V**ideo Reward Feedback via **E**nsemble **L**atent **R**eward Models, a scalable and robust ReFL framework tailored for T2V models, as shown in Fig. 1. Our contributions are threefold:

(1) **Ensemble Latent Reward Models.** We introduce ensemble LRMs that enhance spatio-temporal expressiveness, provide uncertainty estimation, and mitigat reward hacking. This design significantly reduces memory use and speeds up training, making ReFL feasible for sota T2V models.

(2) **Efficient ReFL Training Strategy.** We develop a training procedure tailored for the VELR framework with two key components: a truncated mid-step setting that enables fast and effective gradient updates, and an online alignment mechanism that keeps reward predictions calibrated to ground-truth rewards.

(3) Together, these techniques reduce memory usage by up to **87.6%**, enabling ReFL with large-scale video reward models (up to 32B parameters) on sota T2V models, including OpenSora1.2 (Zheng et al., 2024), CogVideoX1.5(Hong et al., 2023; Yang et al., 2025), and Wan-2.1 (Wan et al., 2025). Extensive experiments demonstrate that VELR is efficient, scalable, and robust, making ReFL feasible at previously unattainable model and reward scales. We provide video comparisons on our project website.

**Conflict of Interest Disclosure**. The authors declare no financial conflicts of interest. All models and code used in this work are based on publicly available open-source resources. No proprietary models developed by any author's employer were evaluated, and no financial interests are involved in the research presented in this paper.

## 2. Related Work

Reinforcement learning has become a standard paradigm for aligning diffusion-based video generation models with human preferences. Existing RLHF methods can be roughly categorized into three groups. Policy-gradient-based ap-

proaches, such as DDPO (Black et al., 2024), optimize the diffusion model as a stochastic policy but often suffer from instability and sample inefficiency. Group-based policy optimization methods, including DanceGRPO (Xue et al., 2025) and FlowGRPO (Liu et al., 2025a), estimate the advantage function via a group of samples, which substantially increases inference cost. Preference-optimization methods, such as DPO (Wallace et al., 2024; Zhang et al., 2024a; Wu et al., 2025c), construct paired samples to reflect human choices, but rely on large-scale curated preference datasets. Overall, while RLHF provides a principled alignment mechanism, these approaches are either unstable, computationally demanding, or data-intensive.

ReFL can be regarded as a variant of RLHF, which enables faster optimization and yields competitive performance (Xu et al., 2023). ReFL has already demonstrated strong performance in the image generation domain. (Chen et al., 2024a) incorporated text-encoder feedback to strengthen semantic alignment, (Fan et al., 2024) adapted ReFL for prompt tuning to enhance abstract concept understanding, Hyper-SD proposed a trajectory-segmented consistency model, integrating ReFL to accelerate image synthesis (Ren et al., 2024), unified multiple feedback signals to improve latent diffusion training (Zhang et al., 2024b), and ImageReFL applied ReFL to balance quality and diversity in human-aligned diffusion models (Sorokin et al., 2025). A recent work Dollar explores the use of a latent reward model for ReFL (Ding et al., 2025). However, it focuses solely on fine-tuning consistency models with image-based and reward models with relatively simple architectures and limited representational capacity.

In the context of video generation, applying ReFL is challenging due to its high memory requirements and the complex design of the VAE decoder in T2V models (Hong et al., 2023). As a result, existing works have only explored ReFL with image-based reward models, such as InstructVideo(Yuan et al., 2024) and VADER (Prabhudesai et al., 2025). In contrast to these works, VELR is the first to apply the ReFL framework on T2V models using **large-scale video RMs**, and the latent reward model used in VELR significantly outperforms the one proposed by Dollar.

## 3. Preliminaries

### 3.1. Diffusion Model

The diffusion models aim to approximate the data distribution $x_0 \sim q(x_0)$ by defining a stochastic process that gradually perturbs the data with Gaussian noise and then learns to reverse this process in order to recover samples from $q(x_0)$ (Rombach et al., 2022).

The forward process in the latent space is defined as a Markov chain, where a clean latent variable $z_0$ is progres-

sively corrupted into $z_t$ through a predefined noise schedule $\alpha_t, t \in [1, \cdots, T]$ (Ho et al., 2020; Song et al., 2020):

$$z_t = \sqrt{\bar{\alpha}_t} z_0 + \sqrt{1 - \bar{\alpha}_t}, \epsilon, \quad \epsilon \sim \mathcal{N}(\mathbf{0}, \mathbf{I}), \quad (1)$$

where $\bar{\alpha}_t = \prod i = 1^t \alpha_i$ and $\epsilon \sim \mathcal{N}(\mathbf{0}, \mathbf{1})$ represents the normal Gaussian noise.

The reverse process seeks to recover $z_0$ from $z_t$ by learning a neural approximation of the injected noise in diffusion models. In the standard velocity parameterization, the model $v_\theta$ is trained to predict the ground-truth velocity $v_t$, leading to the following objective:

$$L_V(\theta) = \mathbb{E}_{x_0 \sim q(x_0), \epsilon \sim \mathcal{N}(0, I), t}[\omega_t || v_\theta(x_t, t) - v_t ||_2^2]. \quad (2)$$

For rectified flow (Liu et al., 2022), the interpolation between clean data $x_0$ and noise $\epsilon$ is defined by coefficients $a_t = 1 - t, b_t = t$ with $t \in [0, 1]$. In this case, the velocity field becomes constant.

### 3.2. Reward Feedback Learning

ReFL (Xu et al., 2023) is a reinforcement learning paradigm that leverages pretrained reward models to guide policy optimization. Given a reward model $R_p(v, c)$ trained to predict human or learned preferences over generated outputs, the latent diffusion model is considered as a policy $\pi_\theta(z_0|c)$. the ReFL updates the policy $\pi_\theta$ by maximizing the expected reward on a dataset $\mathcal{D}$ composed of prompts:

$$\theta^* = \arg\max_\theta \mathbb{E}_{c \sim \mathcal{D}, v \sim \pi_\theta(z_0|c)}[R_p(v, c)]. \quad (3)$$

## 4. Methodology

To mitigate the excessive memory cost of video RMs while leveraging their superior capability, we adopt a latent reward modeling strategy. Sec. 4.1 presents the architecture of the proposed latent reward model (LRM) and its training paradigm on a mixed dataset of real and generated videos. In Sec. 4.2, We enhance the LRM by introducing ensemble-based technique to facilitate robust performance, uncertainty estimation and conservative update. Sec. 4.3 details the ReFL fine-tuning paradigm, with two key components: truncated mid-step setting and online alignment of LRM. The overall architecture of VELR is illustrated in Fig 2.

### 4.1. Architecture and Training Paradigm of LRM

**Network**. We first introduce the network architecture of LRM $\mathcal{R}_l(\boldsymbol{Z}^v, \boldsymbol{Z}^c) : \mathbb{R}^{C_l \times T_l \times H_l \times W_l} \times \mathbb{R}^{d_t \times d_c} \to \mathbb{R}$, with $\boldsymbol{Z}^v$ representing the latent variable $z_0$ in matrix form, and $\boldsymbol{Z}^c$ denotes the text embedding of the prompt $c$. The LRM architecture combines 3D residual convolutions with a Transformer encoder to predict scalar rewards from latent

representations. The design focuses on the local spatiotemporal structure with the capacity to capture long-range dependencies, providing a compact yet expressive architecture. Besides, the textual features are extracted from the prompt by the text encoder and the prompt embeddings are injected into the video features by cross-attention modules to ensure text-video alignment.

To be specific, videos $\boldsymbol{X}^v \in \mathbb{R}^{C \times T \times H \times W}$ are fed to the VAE encoder of the diffusion model to obtain latent representations $\boldsymbol{Z}^v \in \mathbb{R}^{C_l \times T_l \times H_l \times W_l}$. The latent representations are first processed by a stack of 3D residual blocks to obtain the video features $f_v$. Each block applies two 3D convolutions with group normalization and residual connection:

$$\boldsymbol{H}^{(l)} = f\big(f(\boldsymbol{H}^{(l-1)}; \boldsymbol{W}_1^{(l)}); \boldsymbol{W}_2^{(l)}\big) + \boldsymbol{D}(\boldsymbol{H}^{(l-1)}), \quad (4)$$

where $f(\boldsymbol{X}; \boldsymbol{W}) = \sigma\big(GN(\boldsymbol{W} * \boldsymbol{X})\big)$ represents a projection layer, $l \in \{1, 2, 3\}$ represents the list of blocks, $\boldsymbol{H}^{(0)} = \boldsymbol{Z}^v$, $\boldsymbol{W}_1^{(l)}$ and $\boldsymbol{W}_2^{(l)}$ represent the convolutional blocks, $*$ denotes 3D convolution, $\sigma$ is SiLU, and $\boldsymbol{D}$ is a downsampling mapping. The resulting feature $\boldsymbol{H}^{(3)}$ is reshaped into a sequence of video features $\boldsymbol{Z}^{vf} \in \mathbb{R}^{(T_l H_l W_l) \times d_v}$, with $d_v$ being the feature dimension. Then the video feature is augmented with temporal and spatial positional embeddings $\boldsymbol{Z}^{vf} = \boldsymbol{Z}^{vf} + \boldsymbol{P}_v^\phi$, with the $\boldsymbol{P}_v^\phi$ being a learnable position embedding vector.

We then employ cross attention to achieve cross-modal alignment between video features and textual prompts (Gorti et al., 2022). Specifically, cross attention allows video representations to selectively aggregate semantic information from the text while preserving their spatiotemporal structure, ensuring consistency with the prompt and avoiding the computational overhead of concatenating modalities. The prompt $p$ is first embedded to text embeddings $\boldsymbol{Z}^c \in \mathbb{R}^{d_t \times d_c}$, with $d_t$ being the number of tokens and $d_c$ being the dimension of prompt features. To keep the network lightweight, we follow CLIP (Radford et al., 2021) and use only the last token of the embedding as input $\boldsymbol{Z}_{d_t,:}^c \in \mathbb{R}^{1 \times d_c}$. This last token captures most of the information while significantly reducing the data dimensionality. The last text feature $\boldsymbol{Z}_{d_t,:}^p$ is projected to the video feature dimension $d_v$ to obtain the key and value in the cross attention module $\boldsymbol{P}_f = \boldsymbol{Z}_{d_t,:}^p \boldsymbol{W}_p$, with $\boldsymbol{W}_p \in \mathbb{R}^{d_p \times d_v}$. The cross attention block is represented as below:

$$\text{Attn}(\boldsymbol{Z}^{vf}, \boldsymbol{Z}_{d_t,:}^c, \boldsymbol{Z}_{d_t,:}^c) = \text{softmax}\big(\frac{\boldsymbol{Z}^{vf}(\boldsymbol{Z}_{d_t,:}^c)^T}{\sqrt{d_v}}\big)\boldsymbol{Z}_{d_t,:}^c. \quad (5)$$

Then the outputs are passed to a multi-head self-attention and feedforward layers. Finally, we aggregate the token representations and predict the scalar reward $R_l^\phi$.

**Offline Training of LRM**. We train the LRM on a dataset $\mathcal{D} = \{\boldsymbol{Z}_i^v, \boldsymbol{Z}_i^c\}_{i=1}^N$. We adopt the dataset of MVQA-

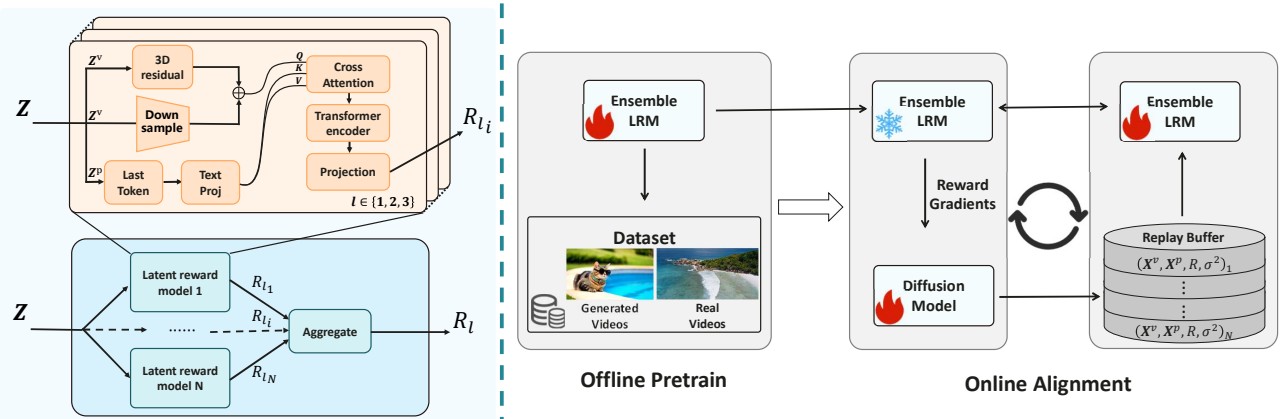

*Figure 2.* **The overall architecture of VELR.** Left: ensemble-based LRM architecture. Right: training procedure, the ensemble-based LRM is first pre-trained on a dataset and then used to guide diffusion model updates.

68K (Pu et al., 2025) which is composed of real high-quality videos, as detailed in Appendix B.1. We further construct a complementary dataset of videos generated by the diffusion model to augment training. The real videos provide rich visual and semantic diversity, which enhances the generalization ability and robustness of the LRM. Meanwhile, the generated samples ensure that the LRM is well aligned with the latent distribution induced by the diffusion model, thereby enabling it to accurately evaluate outputs during video generation. The combination of these two sources of data allows the LRM to simultaneously capture the distributional characteristics of generated samples and retain reliable performance on real data.

We train the latent reward model based on the combined dataset with huber loss. Huber loss combines the benefits of mean square error (MSE) and mean absolute error (MAE): it behaves like MSE for small errors, enabling precise fitting, and like MAE for large errors, making it robust to outliers. Compared to MSE, it stabilizes training and improves convergence, especially on noisy data. Formally, the loss is denoted as $\mathcal{L}_{lrm}^{\delta}(\mathcal{R}(\text{Dec}(\cdot)), R_l^{\phi}(\cdot))$,

$$\mathbb{E}_{\boldsymbol{Z}_i^v, \boldsymbol{Z}_i^p \in \mathcal{D}} \left[ \begin{cases} \frac{1}{2}\left(\Delta(z)\right)^2, & \text{if } |\Delta(z)| \le \delta \\ \delta\left(|\Delta(z)| - \frac{1}{2}\delta\right), & \text{if } |\Delta(z)| > \delta \end{cases} \right], \quad (6)$$

where $\text{Dec}(\cdot)$ represents the VAE decoder and $\Delta(z) = \mathcal{R}(Dec(\boldsymbol{Z}_i^v), \boldsymbol{Z}_i^p) - R_l^{\phi}(\boldsymbol{Z}_i^v, \boldsymbol{Z}_i^p)$.

### 4.2. Ensemble-based LRM

We first observe that the current LRM exhibits unsatisfactory predictive performance: it fails to accurately align with the ground-truth reward model, showing noticeable prediction errors on both the training and test sets (as later confirmed in Table 3). This observation suggests that even within the in-distribution regime, the LRM provides only a coarse and imperfect approximation of the true reward model.

This limitation becomes even more pronounced under distribution shift. As fine-tuning progresses, the distribution of the diffusion model's backbone inevitably shifts; consequently, the generated video latents $z_0$ produced by the model also evolve accordingly. Consequently, the LRM increasingly encounters samples that fall outside the offline distribution, i.e., out-of-distribution (OOD) samples. Notably, these OOD samples include particularly informative cases: such as videos receiving unusually high or low rewards, which are precisely the instances that guide ReFL updates toward desirable or undesirable directions. However, the current LRM generalizes poorly to such important OOD examples, often exhibiting large deviations from the ground-truth reward.

To address these issues, we introduce an **ensemble-based LRM**. By combining multiple models, the ensemble can better capture uncertainty and improve generalization on OOD latents. Ensemble techniques have been explored in offline RL for estimating the Q-function, which can improve the model's ability, mitigate biases introduced by a single network, provide stronger robustness to out of distribution (OOD) data (An et al., 2021; Peer et al., 2021). In general, these methods train multiple models independently and aggregate their predictions to obtain a more reliable estimate. For the LRMs, the ensemble technique is incorporated into the LRM through the following formulation:

$$\mathcal{R}_l(\boldsymbol{Z}^v, \boldsymbol{Z}^p) = \text{Agg}(\mathcal{R}_1(\boldsymbol{Z}^v, \boldsymbol{Z}^p), ..., \mathcal{R}_N(\boldsymbol{Z}^v, \boldsymbol{Z}^p)),$$
$$(7)$$

where $\mathcal{R}_i(\boldsymbol{Z}^v, \boldsymbol{Z}^p)(i \in 1, ..., N)$ represents the reward function trained independently, whereas Agg denotes an aggregation operator, which may be instantiated as the maximum, minimum, or mean operator. The ensemble tech-

niques can enhance the expressive capacity of the learned LRM and improve robustness to OOD samples through collective predictions. Furthermore, the variance among ensemble members provides a natural estimate of prediction uncertainty: samples with high variance are more likely to lie outside the training distribution and therefore may require further alignment with the ground-truth reward. Moreover, given the relatively lightweight architecture of the LRM, employing ensemble techniques does not introduce significant computational overhead.

### 4.3. Training Paradigm of VELR

After pretraining the ensemble-based LRM, we leverage it to fine-tune the T2V model using the ReFL algorithm. For the loss function, we incorporate a KL divergence regularization term to mitigate reward hacking:

$$\mathcal{L} = -\mathbb{E}_{c\sim\mathcal{D}_c, z_0\sim\pi_\theta(z_0|c)} \left[ R_l(z_0, c) - \beta KL\left[\pi_\theta, \pi_{\theta_{old}}\right]\right], \tag{8}$$

where $\mathcal{D}_c$ represents the dataset of prompts and $\pi_{\theta_{old}}$ represents the initial model.

**Truncated Mid Step Setting**. We observe that updating the model only at the late, low-noise denoising steps has minimal effect on video generation, which makes the ReFL solution in VADER and ImageReward ineffective. This is because the sampling trajectory of the video generative model quickly enters a low-noise regime where additional perturbations have negligible impact. For instance, in Wan-2.1-1.3B, adding noise after 30 denoising steps out of 50 produces almost no observable change (see Appendix D for results). Conversely, at early denoising steps, the video reward model exhibits limited discriminative ability: latents derived directly from the velocity field remain highly blurred and provide unreliable feedback. Taken together, these observations indicate that fine-tuning solely at the terminal low-noise stages or at the very early velocity-derived latents is ineffective. Motivated by this analysis, we propose a *truncated mid-step* setting: we randomly select a denoising step in the mid noise regime that has a significant impact on the video as the starting point for retaining gradients, then denoise for several subsequent steps until frames become relatively clear, after which velocity-field guidance is used to generate the final video. This strategy enables effective fine-tuning in mid-noise steps while mitigating the adverse effects of overly blurry generations. This paradigm also effectively reduces training time, as the number of denoising steps involved in training is significantly decreased. The training procedure is illustrated in Fig. 1.

**Online Alignment of LRM**. Empirically, we observe that without updating the LRM, the discrepancy between the latent and true rewards grows steadily throughout fine-tuning, indicating its limited generalization to OOD regions (Refer to Appendix B.7). This observation underscores the neces-

sity of enabling the LRM to not only evaluate OOD samples but also maintain alignment with the ground-truth reward during ReFL training.

To this end, we maintain a replay buffer $\mathcal{B} = \{(X^v, X^p, R, \sigma^2)\}_N$, which stores the inputs required by the ensemble-based LRM, the corresponding pixel-space reward, and the variance of the LRM outputs. During diffusion model updates, new samples are inserted into the replay buffer. The buffer is organized according to the variance $\sigma^2$, and samples with higher variance replace those with lower variance, ensuring that more informative samples are retained. At fixed intervals during training, the LRM is further fine-tuned using the samples stored in the replay buffer. To prioritize informative data, we employ the variance magnitude $\sigma^2$ as the sampling weight, assigning higher probabilities to samples with greater uncertainty and thereby improving the accuracy of reward modeling. To control memory usage and maintain computational efficiency, we adopt a small-scale replay buffer, which our experiments demonstrate to be sufficient in practice.

## 5. Results

In the experiments, we focus on the following aspects: (1) To what extent can VELR reduce memory consumption compared to standard ReFL algorithms? (2) Does VELR maintain performance comparable to standard ReFL despite the memory reduction? (3) Can VELR remain effective in scenarios where standard ReFL are computationally infeasible? (4) What are the contributions of the individual components of VELR to the overall effectiveness?

**Baselines.** We compare against the following methods:

- **Base models**. OpenSora 1.2 (Zheng et al., 2024) CogVideoX-1.5 (Hong et al., 2023; Yang et al., 2025) and Wan-2.1 (Wan et al., 2025) are current state-of-the-art open-sourced T2V diffusion models. We consider them as base models for ReFL and VELR.

- **ReFL solutions**. VADER (Prabhudesai et al., 2025) and ImageReward (Xu et al., 2023) are state-of-the-art ReFL solutions. We adopt their updating paradigm, including truncated steps and feedback steps.

- **LRM-based ReFL**. Dollar (Ding et al., 2025) is the first ReFL solution based on LRMs on the consistency model. Though it is not open sourced, we believe it is an important baseline and we reproduce it based on its description.

**Reward models.** We utilize the following RMs.

- **PickScore** (Kirstain et al., 2023) is a preference model

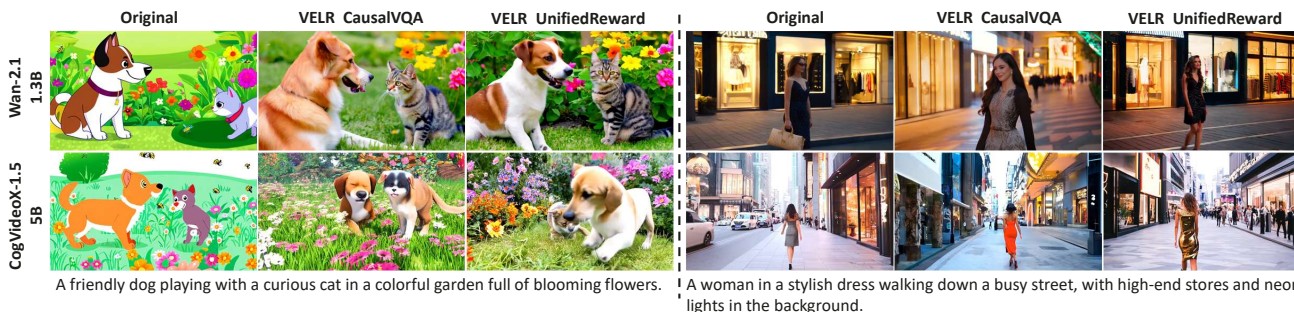

*Figure 3.* Results of VELR on Wan-2.1 and CogVideoX-1.5. Each pair of rows corresponds to the same input: the first row shows results of Wan-2.1, and the second row of CogVideoX-1.5. Additional videos results are analysed in Appendix C. **Whole video results are provided in the website**.

*Table 1.* Memory consumption (GB) of different modules during gradient backpropagation under the UnifiedReward setting.

| Method | RM/LRM | Decoder | DiT | Total |
|---|---|---|---|---|
| ReFL | 105.44 | 54.76 | 18.39 | 178.59 |
| VELR | 4.79 | / | 18.39 | 22.18 |
| Reduction | 100.65 | 54.76 | 0.00 | **156.41** |

trained on human pairwise comparisons, serving as a scalable evaluation metric for text-to-image generation.

- **CausalVQA** (Pu et al., 2025) is built on VideoAlign (Liu et al., 2025b) and fine-tuned on a diverse set of high-quality datasets, resulting in enhanced capabilities.

- **UnifiedReward** (Wang et al., 2025b). We adopt UnifiedReward as one of our reward models and utilize its pointwise scoring functionality to provide fine-grained, human-aligned evaluation for video generation.

### 5.1. Memory Optimization

To illustrate the substantial memory consumption of the standard ReFL algorithm and the memory efficiency of VELR framework, we conduct a detailed memory analysis with batch size set to 1 under the Wan-2.1 setting. For ReFL, multiple memory optimization techniques are required, as detailed in Appendix B.3, reflecting significant practical compromises. In particular, the video RMs in ReFL can only process a single video frame at a time, while VELR directly feeds the entire latents into the LRMs without any such restriction. Table 1 and 6 (In the Appendix) report the memory usage of different modules. On average, the memory overload is reduced to **16.7%** of that required by the ReFL solution. The results demonstrate that VELR substantially reduces memory consumption while preserving full-sequence modeling ability.

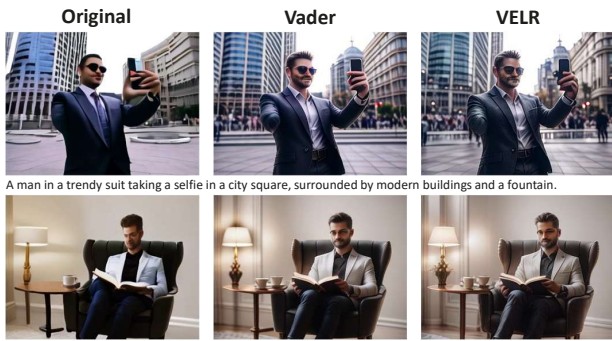

*Figure 4.* Comparison of VELR and VADER on Image RM PickScore. The first, middle, and last columns correspond to the videos generated by the original model, VADER, and VELR, respectively.

### 5.2. VELR with Image Reward Models

To verify that VELR can achieve performance comparable to standard ReFL while using LRM estimation with lower memory consumption, we first conduct experiments with image reward models, where the standard ReFL algorithm is **applicable**. We train both ReFL and VELR on OpenSora-1.2 using the PickScore-v1 RM (Kirstain et al., 2023). Notably, VELR reduces memory usage from 38.89GB to 19.73GB, achieving a **50% reduction**. Fig. 4 reports the fine-tuning results. The results demonstrate that, despite the substantial memory reduction, VELR evolves in the same direction as standard ReFL, indicating that LRM module effectively aligns with the original reward model. In addition, VELR produces richer visual details and higher video quality, suggesting faster performance improvement under the same number of training steps.

### 5.3. VELR with Video Reward Models

Furthermore, we extend the VELR paradigm to settings where the standard ReFL is infeasible. This extension allows

*Table 2.* Evaluation results across multiple metrics from Vbench. The optimal results are **bolded**.

| Methods | Overall Consistency | Aesthetic Quality | Human Fidelity | Composition | Image Quality | Average |
|---|---|---|---|---|---|---|
| Wan-2.1 1.3B | 22.89 | 64.01 | 81.76 | 38.15 | 66.41 | 54.64 |
| VELR Causal-VQA | 23.32 ↑ +0.43 | 64.74 ↑ 0.63 | 84.37 ↑ +2.61 | 41.79 ↑ +3.64 | 64.54 ↑ +0.53 | 55.55 ↑ +0.91 |
| VELR unified-reward | 23.35 ↑ +0.46 | **66.21** ↑ 2.20 | **86.85** ↑ +5.09 | 43.98 ↑ +5.83 | **69.62** ↑ +3.21 | **58.01** ↑ +3.37 |
| CogVideoX-1.5 5B | 27.71 | 62.53 | 61.92 | 43.85 | 65.26 | 52.25 |
| VELR Causal-VQA | 28.14 ↑ +0.43 | 61.94 ↓ -0.59 | 68.59 ↑ +6.67 | 46.29 ↑ +2.44 | 65.18 ↓ -0.06 | 54.03 ↑ +1.78 |
| VELR unified-reward | **31.26** ↑ +3.55 | 65.18 ↑ +2.65 | 71.04 ↑ +9.12 | **51.43** ↑ +12.28 | 66.87 ↑ +0.46 | 57.16 ↑ +4.91 |

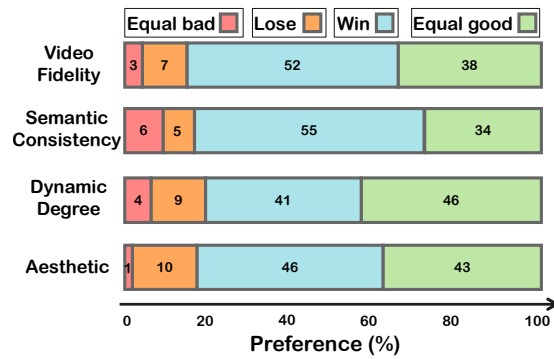

*Figure 5.* Human evaluation results. The blue portion (win) is significantly higher than the orange portion (lose).

*Table 3.* The comparison of different LRM configurations.

| LRM | Dollar | | LRM-en-1 | | LRM-en-5 | | LRM-en-10 (ours) | |
|---|---|---|---|---|---|---|---|---|
| | MSE | PCC | MSE | PCC | MSE | PCC | MSE | PCC |
| Train set | 0.832 | 0.59 | 0.315 | 0.79 | 0.072 | 0.92 | **0.003** | **0.98** |
| Test set | 1.215 | 0.34 | 0.851 | 0.68 | 0.158 | 0.84 | **0.008** | **0.91** |

ble 2, show that VELR consistently outperforms the original models, achieving an average improvement of **2.74**, which demonstrates its superior generalization. Additional details regarding the evaluation protocols and metric computation are provided in Appendix B.5.

### 5.5. Human Evaluation

To validate VELR's alignment with human preferences, we carried out a study to evaluate human preferences. We include dimension about Video Fidelity, Semantic Consistency, Dynamic Degree and Aesthetic. Qualitative results, illustrated in Fig. 5, shows a low lose rate (3.5%) and a substantially higher win rate (48.5%), which supports that VELR consistently outperforms the original models. For detailed protocols of the human evaluation process, please refer to Appendix B.6.

### 5.6. Ablation Study

**Truncated mid-step setting**. We conducted extensive ablation studies on the two key hyperparameters of the truncated mid-step setting: the denoising starting step $N$ at which gradients start to be retained, and the truncated duration $K$. Each variant is denoted as VELR-N-K, and the overall results are shown in Fig. 6.

For the denoising starting step $N$, we observe that choosing an early step leads to blurry outputs, as the T2V model is not yet capable of producing clear frames, and these blurry intermediate predictions ultimately guide the model toward blurry final videos. Conversely, setting $N$ too late results in minimal changes, consistent with the phenomenon discussed in Sec. 4.3: gradients from late steps have limited influence on the video content.

more powerful diffusion models with larger and stronger reward models. The qualitative results are shown in Fig. 3. While the standard ReFL becomes impractical under advanced video diffusion models, VELR continues to operate effectively and yields encouraging results. Compared to the base models, VELR fine-tuned outputs exhibit more realistic video quality and significantly stronger text–video alignment. For example, in the left prompt, the baseline Wan-2.1 produces blurry video details and CogVideoX-1.5 generates irrelevant and abnormal cars, while the VELR-fine-tuned results exhibit clearer details of the woman and a street scene more consistent with the prompt.

Furthermore, given that our setup optimizes video diffusion models based on video RMs, temporal consistency is a meaningful evaluation dimension. VELR demonstrates strong performance in this regard, reducing artifacts and flickering while maintaining consistently high-quality subjects. Detailed comparisons are provided in Appendix C and refer to our website for additional video results.

### 5.4. Automatic Evaluation on VBench

To evaluate the effectiveness of VELR on out-of-distribution metrics, we compare the fine-tuned and original models on the VBench (Huang et al., 2024) and VBench2.0 (Zheng et al., 2025) benchmarks. The results, summarized in Ta-

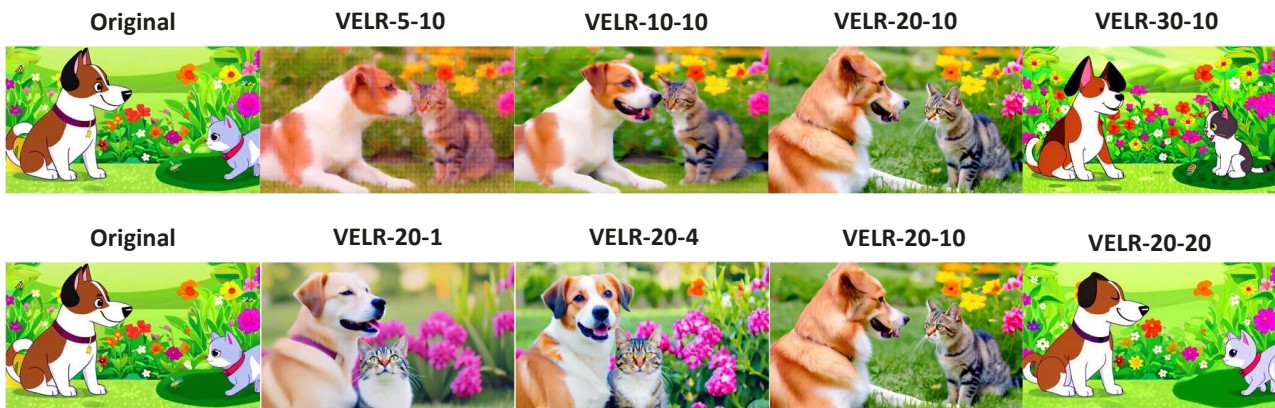

*Figure 6.* Ablation study on the truncated mid-step setting.

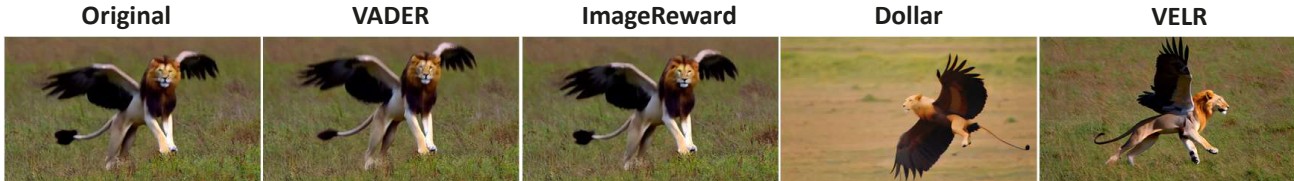

A lion with the wings of an eagle, soaring through the sky with majestic ease.

*Figure 7.* Results of different ReFL update paradigm.

For the truncated length $K$, a small $K$ behaves similarly to an early $N$, producing blurry videos and unnatural lighting. A large $K$ accumulates gradients from many late denoising steps, but these late-step gradients dominate while carrying little meaningful information, diluting the earlier informative gradients and resulting in minimal updates to the model.

Furthermore, we compared results within the 15–25 step range (see Appendix B.8) shows largely similar results, indicating that the truncated mid-step setting is **insensitive** to the selection of effective intervals. We apply the same hyperparameters across all models and reward models (see Appendix B.1), demonstrating the usability and robustness of the VELR paradigm.

To further support the truncated mid-step setting, we provide additional quantitative comparisons on VBench against relevant variants. VELR consistently achieves the best performance across all metrics, providing empirical support for the chosen N and K values. **Effectiveness of**

*Table 4.* Comparison of different VELR configurations on the truncated mid-step setting. The optimal results are **bolded**.

| Metric | wan-2.1 | VELR | VELR-10-10 | VELR-20-4 | VELR-30-10 |
|---|---|---|---|---|---|
| Overall Consistency | 22.89 | **23.35** | 23.02 | 22.87 | 22.91 |
| Subject Consistency | 95.12 | **95.96** | 95.35 | 95.36 | 95.08 |
| Aesthetic Quality | 64.01 | **66.21** | 65.26 | 64.35 | 64.05 |
| Image Quality | 66.41 | **69.62** | 67.34 | 66.90 | 66.39 |

**ensemble-based LRM**. We investigate different LRM configurations, including the LRM proposed in Dollar (Ding et al., 2025), the non-ensemble variant described in Section 4.1, and ensemble-based LRM models with varying numbers of components (3, 5, and 10). All models are trained under identical settings, and their performance on both training and test sets is reported in Table 3. The results clearly demonstrate that ensemble-based LRM significantly outperforms Dollar. Furthermore, ensemble methods substantially enhance the representational capacity of latent reward models. The superior test performance also highlights that ensembles improve generalization and strengthen robustness against OOD samples. For more ablation results on the LRM components and OOD generalization, see Appendix B.8.

**Training Paradigms**: We further compare the impact of different update paradigms on Wan-2.1 model. Specifically, we consider the intermediate-step sampling paradigm proposed by ImageReward (Xu et al., 2023), the truncated-last-N-steps paradigm represented by Vader (Prabhudesai et al., 2025). Because of memory overload, we implement the LRM version of these paradigms. We also include the LRM-based baseline Dollar (Ding et al., 2025). All paradigms are trained with identical settings. The results are shown in Fig. 7. It can be observed that the first two approaches have limited influence on the intrinsic attributes of the generated

| Original | VELR w/o OA | VELR w/o UW | VELR |
|---|---|---|---|

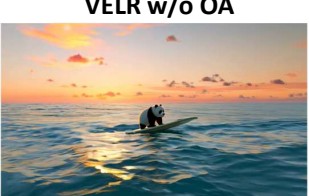

A panda standing on a surfboard in the ocean in sunset.

*Figure 8.* Ablation on online alignment and uncertainty weighting mechanisms.

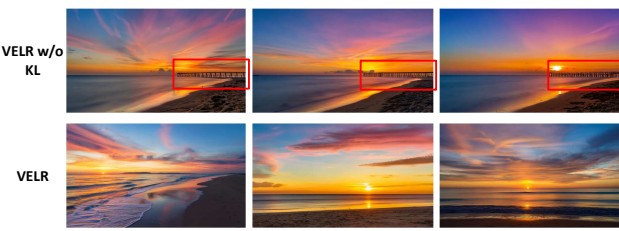

Sunset time lapse at the beach with moving clouds and colors in the sky.

*Figure 9.* Ablation on the KL divergence term, the variant is denoted as VELR w/o KL. The videos in each column are generated using the same random seed.

videos. Dollar exhibits blurry background and disorganized subjects. In contrast, VELR provides effective updates, with noticeable improvement in video quality, fidelity, lighting and shadows. This demonstrates the superiority of the VELR update paradigm, showing that VELR can fine-tune the model effectively and robustly.

**KL Divergence Term**: We provide an additional variant without the KL term, VELR w/o KL, and include visual comparisons in Fig.9. The results show that removing the KL term leads to a decrease in generation diversity. However, the generated videos by VELR w/o KL remain visually coherent and do not exhibit the severe artifacts typically associated with reward hacking, confirming the ensemble LRM is the primary mechanism against reward hacking. This is also consistent with findings in prior work (Liu et al., 2025a; Wu et al., 2025a).

**Key Training Mechanisms**: We include variants without online alignment (VELR w/o OA) and without uncertainty weighting (VELR w/o UW). Refer to Fig. 8 for qualitative comparisons and Table 5 for quantitative results.

Compared to the base model, VELR w/o OA shows improved performance, while remaining slightly below VELR on most metrics. We observe that it converges rapidly toward the LRM and exhibits a mild degradation in final quality. This suggests that, without online alignment, the LRM may become less effective when handling OOD samples.

VELR w/o UW demonstrates comparable but slightly weaker performance than VELR in both quantitative met-

*Table 5.* Ablation study of different VELR training mechanisms.

| Metric | wan-2.1 | VELR | VELR w/o OA | VELR w/o UW |
|---|---|---|---|---|
| Overall Consistency | 22.89 | **23.35** | 23.15 | 23.31 |
| Subject Consistency | 95.12 | 95.96 | 95.79 | **95.97** |
| Aesthetic Quality | 64.01 | 66.21 | **66.34** | 66.20 |
| Image Quality | 66.41 | **69.62** | 68.36 | 69.58 |
| Cost (GPU hours) | / | 12.6 | **10.2** | 15.0 |

rics and visual quality, along with a slower convergence trend. This indicates that uncertainty weighting contributes to more stable and efficient alignment of the LRM.

## 6. Conclusion

In this paper, we introduce VELR, which leverages ensemble-based latent reward models to address the prohibitive memory demands of video-RM-based ReFL algorithm. By enhancing the capacity of LRMs and incorporating ensemble estimation, VELR achieves efficient, scalable, and robust performance across large-scale T2V models, which is memory infeasible for standard ReFL algorithms. However, VELR can not fully exploit information from earlier denoising steps, which may limit optimization efficiency. In future work, we aim to develop ReFL algorithms that can stably leverage information from all denoising steps, potentially further improving training efficiency and enhancing overall model performance.

## Acknowledgements

We sincerely thank Yanyun Pu and other members of the Central Research Institute team at Huawei, as well as the anonymous reviewers for their valuable feedback on the paper. This work was done during an internship at Huawei and was supported by Key Research and Development Program of Zhejiang Province (Grant No: 2025C01061).

## Impact Statement

This presents work whose goal is to advance the field of Machine Learning. There are many potential societal consequences of our work, none which we feel must be specifically highlighted here.

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

## A. Pseudo Code

The pseudo code of the training pipeline of VELR is illustrated in Algorithm 1.

---

**Algorithm 1** Training Pipeline

---

**Require:** text-video offline dataset $\mathcal{D}_{off}$, prompt dataset $\mathcal{D}_p$, initial diffusion model parameter $\theta$, initial LRM parameter $\phi$, learning rate $\eta$, pre-train learning rate $\eta_{lrm}$, ODE solver $\Psi$, noise schedule $\alpha(t), \beta(t)$, VAE encoder Enc, decoder Dec, pixel-space RM $\mathcal{R}$, latent reward model $\mathcal{R}_l^\phi$, LRM training threshold $\delta$, tuncated step $K$, replay buffer $\mathcal{B}$.
  // Pre-train ensemble-based LRM
  **repeat**
    Sample $(\boldsymbol{X^v}, \boldsymbol{c}) \sim \mathcal{D}_{off}$
    $R = R(\boldsymbol{X^v}), R_l = R_l(\text{Enc}(\boldsymbol{X^v}), \boldsymbol{c})$
    compute $\mathcal{L}_{lrm}^\delta$ as Eqn. 6
    $\phi \leftarrow \phi - \eta_{lrm}\nabla_\phi \mathcal{L}_{lrm}^\delta$
  **until** convergence
  // Alternately update LRM $\mathcal{R}_l^\phi$ and diffusion model $\theta$
  $\theta^- \leftarrow \theta$
  **repeat**
    **if** upodate Diffusion Model **then**
      // update diffusion model with LRM
      Sample $\boldsymbol{c} \sim \mathcal{D}_p, n \sim \mathcal{N}[\boldsymbol{0}, \mathbf{I}]$
      **for** $i = T, T-1, \ldots, t_{mid}$ **do**
        Predict noise $\epsilon_\theta(z_i, i)$ {no gradient}
        Compute denoised state $\hat{z}_i = f(z_i, \epsilon_\theta)$ {no gradient}
        Update latent with solver: $z_{i-1} \leftarrow \Psi(z_i, \hat{z}_i, i)$ {no gradient}
      **end for**
      $t_{end} \leftarrow t_{mid} - K$
      **for** $i = t_{mid} - 1, \ldots, t_{end}$ **do**
        Predict noise $\epsilon_\theta(z_i, i)$ {with gradient}
        Compute denoised state $\hat{z}_i = f(z_i, \epsilon_\theta)$ {with gradient}
        Update latent: $z_{i-1} \leftarrow \Psi(z_i, \hat{z}_i, i)$ {with gradient}
      **end for**
      $\mathcal{L}_{KL} = KL[\epsilon_\theta(z_{t_{end}}, t_{end}), \epsilon_{\theta^-}(z_{t_{end}}, t_{end})]$ {with gradient}
      Predict the $z_0$ based on the velocity {with gradient}
      compute $\mathcal{L}_{VELR}$ as Eqn. 8{with gradient}
      $\theta \leftarrow \theta - \eta\nabla_\theta \mathcal{L}_{VELR}$ {with gradient}
      Store sample $(X^v, X^p, R, \sigma^2)$ into replay buffer $\mathcal{B}$
    **else**
      // align LRM with $\mathcal{R}$
      Sample $(X^v, X^p, R, \sigma^2) \sim \mathcal{B}$ with weight $\sigma^2$
      $R = R(\boldsymbol{X^v}), R_l = R_l(\text{Enc}(\boldsymbol{X^v}), \boldsymbol{c})$
      compute $\mathcal{L}_{lrm}^\delta$ as Eqn. 6
      $\phi \leftarrow \phi - \eta_{lrm}\nabla_\phi \mathcal{L}_{lrm}^\delta$
    **end if**
  **until** convergence

---

## B. Experiment and Hyperparameter Details

For all the qualitative or pairwise comparisons between different methods, we ensure to use the same random seed.

### B.1. Pre-training of ensemble-based LRM

As outlined in Sec. 4.1, our first step is to pretrain the ensemble-based LRM. We utilize bf16 precision and FlashAttention to improve efficiency.

**Dataset Construction.** For this stage, we adopt the MVQA-68K dataset (Pu et al., 2025), which contains a large collection of high-quality real-world videos. MVQA-68K is curated from several publicly available sources, notably Panda-70M (Chen et al., 2024b) and Koala-36M (Wang et al., 2025a), and covers a broad spectrum of scenarios such as human activities, urban landscapes, wildlife, vegetation, and indoor settings.

Since the videos we fine-tune on are of relatively high resolution (480p for Wan-2.1 and 720p for CogvideoX-1.5), we first filter out all videos in the dataset with a resolution lower than 480p. In addition, our experiments only consider landscape videos, as they generally yield better performance; thus, portrait videos are further removed. After applying these filtering rules, we obtain a final collection of 4.16K videos. It is worth noting that MVQA-68K dataset contains only raw videos without their corresponding prompts. To enable effective training of the latent reward model, and to maintain consistency with existing pixel-space reward models, we annotate this video set with textual prompts. Specifically, we employ the Qwen-VL-7B (Bai et al., 2025) model to generate the annotations, using the prompt template provided below.

Furthermore, to ensure that the latent reward model can accurately evaluate samples generated by diffusion models—i.e., to maintain alignment with pixel-space reward models on generated data—we augment the dataset with corresponding diffusion-generated samples. In total, we produce 500 generated videos generated by Wan-2.1 and CogvideoX-1.5. The prompts used for generation cover a diverse range of categories, including humans, animals, plants, as well as indoor and outdoor scenes. In these prompts, we also include the prompts from the ReFL fine-tuning prompt dataset. This is to ensure that the LRM can produce accurate reward during the initial stage of ReFl fine-tuning.

For the hyperparameters of training of the latent reward model, we train the LRM with batch size 4 (without gradient accumulation) and learning rate 2e-5 for Wan-2,1 and 1e-4 for CogVideoX-1.5. The training process lasts for about 3 epochs to converge, which takes about 18 hours.

### B.2. Alternately Training of LRM and Diffusion Model

For this stage, we adopt an alternating update strategy between the two models. Specifically, after every 10 epochs of diffusion model training, we perform two rounds of LRM alignment. During diffusion model updates, we use a batch size of 4, with a learning rate of 1e-5 for Wan-2.1 and 2e-5 for CogVideoX-1.5. We maintain a replay buffer of size 256 (our experiments show that this size is sufficient, as enlarging the buffer does not lead to further performance improvements). To update the buffer, we replace the samples with the smallest variance using those from the current batch, ensuring that the buffer always stores the most uncertain samples of the model. For the Wan-2.1 model, we set the mid step to be 20 and the truncated step to be 10; for the CogVideoX-1.5 model, we set the mid step to be 20 and the truncated step to be 10.

To mitigate reward hacking, we incorporate a KL divergence regularization term with a weight of 10. As for the LRM alignment step, it largely follows the same setup as the offline training phase, except that we adopt a smaller learning rate, uniformly set to 1e-5.

### B.3. More Detailed Study on the Memory Overlaod

To illustrate the advantages of the VELR on ReFL, we conduct a more detailed memory overload experiments. Under the Wan-2.1 setting, the batch size is restricted to 1. For the ReFL algorithm, we adopt all possible techniques to optimize memory usage. Specifically, we employ LoRA (Hu et al., 2022) fine-tuning with a rank of 256, use bf16 precision, and apply gradient checkpointing to the DiT module, the decoder, and the reward model. In addition, the reward model is offloaded to the CPU when not in use, and the text embeddings of prompts are precomputed in advance, so the embedding module is not loaded during training.

In particular, to further reduce memory consumption, the video reward model in ReFL only takes a single video frame as input. This design means that part of the latent representation must first be decoded by the VAE decoder and then passed into the video reward model. By contrast, our proposed VELR framework feeds the entire latent sequence directly into the latent reward model, without any such compromise.

We provide a detailed breakdown of memory usage across different modules during backpropagation, as shown in Table 1

and we provide the results on CausalVQA in Table 6 as well. The results reveal that, despite the significant compromises and approximations adopted in ReFL—where the reward model only processes a single frame—the gradients of the reward model and the VAE decoder still dominate memory consumption. This results in high infrastructure demands for ReFL. In contrast, VELR effectively eliminates the gradients of these two modules while maintaining competitive performance, thereby demonstrating both its effectiveness and memory efficiency.

*Table 6.* Memory consumption (GB) of different modules during gradient backpropagation under the **CausalVQA** setting.

| Method | RM/LRM | Decoder | DiT | Total |
|--------|--------|---------|-------|-------|
| ReFL | 32.61 | 54.76 | 18.39 | 105.76 |
| VELR | 4.79 | / | 18.39 | 22.18 |
| Reduction | 27.82 | 54.76 | 0.00 | 83.58 (20.97%) |

## B.4. Comparison with Image Reward Model

To further demonstrate the superiority of VELR with a video reward model, we compared the performance of VELR against ReFL using an image reward model. The results, shown in the figure, further highlight the effectiveness and significant contributions of our proposed paradigm.

## B.5. Details of Vbench Evaluation

Details of the dimensions we choose is shown below:

- **Overall Consistency**: Measures the alignment between generated videos and the provided textual prompts, assessing both semantic accuracy and stylistic coherence across the video sequence. Evaluation uses the ViCLIP model to compute similarity between video and text embeddings.

- **Aesthetic Quality**: Evaluates the perceived visual appeal of generated videos, considering color harmony, composition, and overall artistic quality. Frames are scored using a pretrained aesthetic predictor, then aggregated.

- **Human Fidelity**: Assesses the anatomical correctness and temporal consistency of human figures, ensuring realistic appearance and motion. Specialized models detect anomalies and evaluate identity consistency across frames.

- **Composition**: Measures the spatial arrangement and interaction of objects within the video. Evaluated via spatial relationship and multi-object arrangement metrics, ensuring proper placement and logical scene composition.

- **Image Quality**: Evaluates technical quality of individual frames, including sharpness, noise, and exposure. Assessed with an image quality predictor (e.g., MUSIQ) and aggregated for overall video quality.

We strictly follow the evaluation protocols of VBench and VBench2. Specifically, VBench generates five videos per prompt, while VBench2 generates three. All methods are evaluated under the same random seed to ensure a fair comparison.

## B.6. Human Evaluation

We recruited 18 volunteers with extensive experience in image and video data annotation. For each given prompt, the volunteers were presented with two videos: one generated by the original model and the other generated by the model fine-tuned with VELR. They were asked to conduct a pairwise comparison along four evaluation dimensions, defined as follows:

- **Video Fidelity**: the visual quality and realism of the given video, including clarity, artifact presence, and overall coherence.

- **Semantic Consistency**: the degree to which the video content accurately follows the semantics specified in the prompt.

- **Dynamic Degree**: the consistency and naturalness of temporal motion, capturing whether the video exhibits plausible and abnormal dynamics.

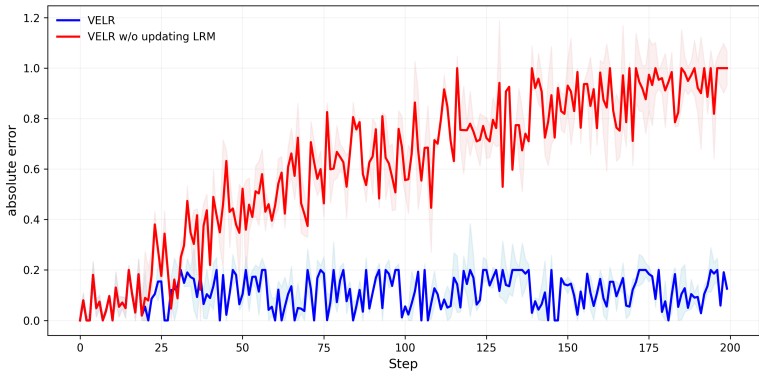

*Figure 10.* **The absolute error curve of the VELR**

- **Aesthetic**: the overall visual attractiveness of the video, reflecting how pleasing and visually appealing it appears to volunteers.

For each dimension, volunteers were instructed to select one of the following four options: *left wins*, *right wins*, *equally good*, or *equally bad*.

Notably, during the experimental design, we observed that both CogVideoX-1.5 and Wan-2.1 are capable of producing high-quality results for a substantial portion of samples. Therefore, instead of using a single tie option, we explicitly distinguish between *equally good* and *equally bad* cases, enabling a more fine-grained and reliable human evaluation.

### B.7. Error Curves

To illustrate the consequences of not updating the LRM, we show the absolute error between the latent rewards and the true rewards during training, as depicted in the fig. 10

### B.8. More Ablation Study on LRM

**LRM Components**. To better illustrate the contribution of each component in the LRM, we conduct ablation studies by evaluating two variants: VELR/C (without the 3D CNN) and VELR/T (without the Transformer). Specifically, VELR/T removes the Transformer Encoder block and directly projects the output of the cross-attention module to the final reward predictions, whereas VELR/C removes the 3D CNN module and downsamples the video features directly to form the $Q$ and $K$ inputs for the cross-attention module. The estimation error and pearson correlation results are shown in table 7 and the generated videos are illustrated in Fig. 11

*Table 7.* **The comparison of different LRM settings on the train and test set.**

| LRM | VELR/T | | VELR/C | | VELR | |
|---|---|---|---|---|---|---|
| | MSE | PCC | MSE | PCC | MSE | PCC |
| Train set | 0.320 | 0.78 | 0.278 | 0.79 | 0.003 | 0.98 |
| Test set | 0.773 | 0.65 | 0.671 | 0.71 | 0.008 | 0.91 |

From the results, we can see that VELR performs substantially better than both variants in quantitative metrics: the variants yield larger MSE and smaller correlation coefficients, demonstrating the superior performance of the ensemble LRM proposed in VELR.

The fine-tuning results further reveal that removing the Transformer module has a severe impact on the LRM. After fine-tuning, VELR/T produces videos that are extremely blurry and exhibit strong flickering. VELR/C removes the 3D CNN module; although it can still improve video quality to some extent, many regions display abnormal details, as highlighted in the red boxes in Fig. 11.

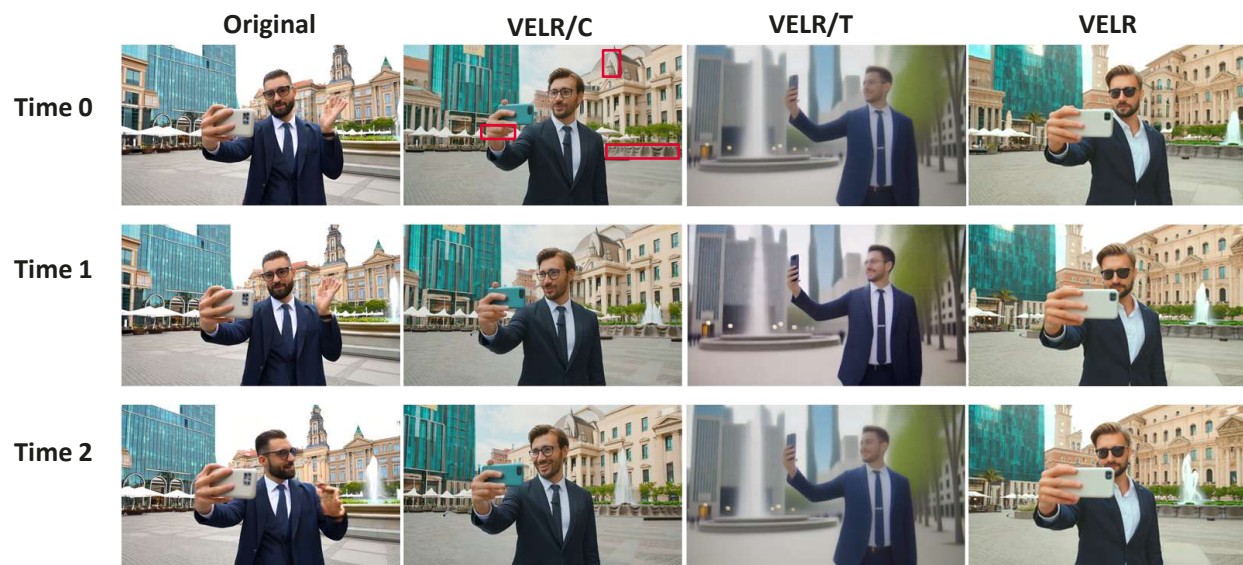

A man in a trendy suit taking a selfie in a city square, surrounded by modern buildings and a fountain.

*Figure 11.* **Video results of different LRM settings. Each row corresponds to the frame at the n-th second of the video, and each column represents a different variant.**

In contrast, the videos fine-tuned with the ensemble LRM show clearly improved and more stable quality. This highlights the importance of both key components within the LRM.

**Long-term consistency of different LRM settings**. To directly assess whether these components in LRM help the LRM learn long-term consistency, we compare metrics related to consistency from VBench. The results are shown in Table 8.

*Table 8.* **Evaluation results across long-term consistency metrics from VBench.**

| Methods | Overall Consistency | Subject Consistency | Background Consistency |
|---|---|---|---|
| Wan-2.1 1.3B | 22.89 | 95.12 | 96.68 |
| VELR/T | 22.46 ↓ -0.43 | 94.56 ↓ -0.56 | 94.69 ↓ -1.99 |
| VELR/C | 22.92 ↑ +0.03 | 95.28 ↑ 0.16 | 96.80 ↑ +0.12 |
| VELR | 23.35 ↑ +0.56 | 95.96 ↑ +0.84 | 97.27 ↑ +0.59 |

Results show that VELR/T underperforms the base T2V model, while VELR/C yields only marginal gains. In contrast, VELR consistently achieves the best performance, demonstrating that the ensemble LRM architecture is essential for modeling long-term consistency.

**LRM Performance on an OOD dataset**. To better demonstrate the robustness of the ensemble LRM architecture against OOD samples, we evaluate the performance of different variants of LRM on an OOD dataset. Specifically, we choose the test set of Panda-70M (Chen et al., 2024b) as the OOD dataset, which contains 6,000 samples. The MSE error and Pearson correlation of different LRM versions on this dataset are reported in Table 9.

As shown, the ensemble LRM retains good predictive ability on OOD samples, while other variants degrade substantially in both correlation and error metrics. This confirms the effectiveness of the ensemble design for improving OOD generalization.

**B.9. Ablation on the Diversity of the Pre-training Dataset**

To assess the sensitivity to pre-training data diversity, we conducted experiments using 50% (VELR-data-50) and 10% (VELR-data-10) of the full dataset, as well as a real-video-only subset (VELR-data-real, around 70% data) to simulate a significantly out-of-domain initialization. Results are shown in Table 10:

*Table 9.* **The comparison of different LRM configurations on an OOD dataset.**

| LRM | Dollar | | LRM-en-1 | | LRM-en-5 | | LRM-T | | LRM-C | | LRM-VELR | |
|---|---|---|---|---|---|---|---|---|---|---|---|---|
| | MSE | PCC | MSE | PCC | MSE | PCC | MSE | PCC | MSE | PCC | MSE | PCC |
| OOD Test set | 1.478 | 0.31 | 0.964 | 0.42 | 0.490 | 0.49 | 0.410 | 0.57 | 0.387 | 0.62 | **0.249** | **0.72** |

*Table 10.* Comparison of different training data configurations.

| Metric | wan-2.1 1.3B | VELR | VELR-data-50 | VELR-data-10 | VELR-data-real |
|---|---|---|---|---|---|
| Overall Consistency | 22.89 | **23.35** | 23.09 | 22.67 | 23.12 |
| Subject Consistency | 95.12 | **95.96** | 95.24 | 95.18 | 95.38 |
| Aesthetic Quality | 64.01 | **66.21** | 65.42 | 64.68 | 65.35 |
| Image Quality | 66.41 | **69.62** | 68.15 | 67.02 | 68.26 |
| Training Time (GPU hour) | / | 12.6 | 14.1 | **9.3** | 19.3 |

These results demonstrate that VELR is not sensitive to pre-training data diversity under reasonable conditions. Both VELR-data-50 and VELR-data-real preserve most of the alignment improvements, even with reduced data volume or out-of-domain initialization. Noticeable performance degradation appears only in the extreme case of using 10% of the data, which corresponds to a severely constrained pre-training setting. In addition, out-of-domain initialization leads to higher convergence cost, suggesting that data diversity mainl y influences training efficiency rather than final performance.

### B.10. Sensitivity analysis of truncated mid step setting

We conducted a sensitivity analysis of the truncated mid step setting and found that selecting the denoising step within the range of 15–25 yields similar results, demonstrating that the truncated mid-step setting is robust to the choice of step number, as shown in Fig. 12.

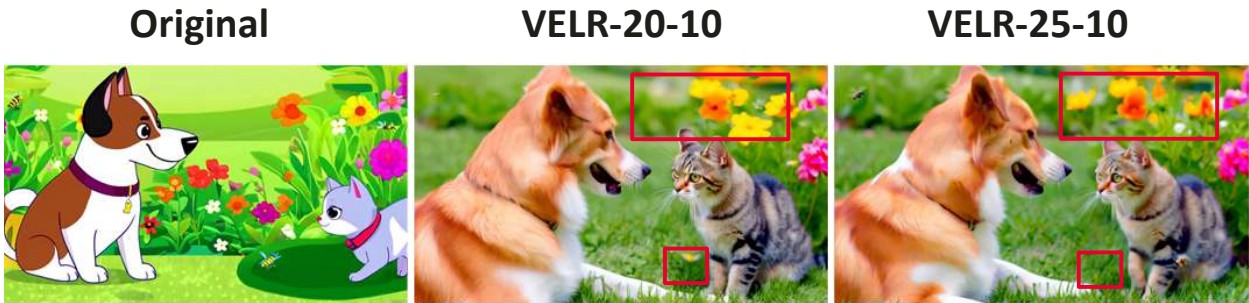

*Figure 12.* **Sensitivity results of VELR.**

The results show that the choice of intermediate steps has only a minor impact on the final outputs. Aside from subtle differences—such as the shape of flowers in the background or fine-grained details like the bee in the foreground or the dog's fur—the other regions remain highly similar. This demonstrates the robustness of VELR to the selection of intermediate steps.

### B.11. Model sizes of Reward Models

The model scales of all reward models used in our work, including PickScore, Causal VQA, and Unified Reward, as well as the latent reward models, are summarized in Table 11.

*Table 11.* **The comparison of different LRM configurations on an OOD dataset.**

|  | PickScore | CausalVQA | UnifiedReward | EVLR(single) | EVLR(ensemble) |
|---|---|---|---|---|---|
| Parameters Size | 1B | 7B | 32B | 0.02B | 0.21B |

## C. Experiment Results

We list more results in the validation set as illustrated in Fig. 13–16. Each pair of rows corresponds to the same prompt: the first row shows results from the original model, and the second row from VELR. Each row contains five frames from a video, evenly spaced in time. As shown in the figure, VELR significantly improves the plausibility of the videos and their alignment with the text.

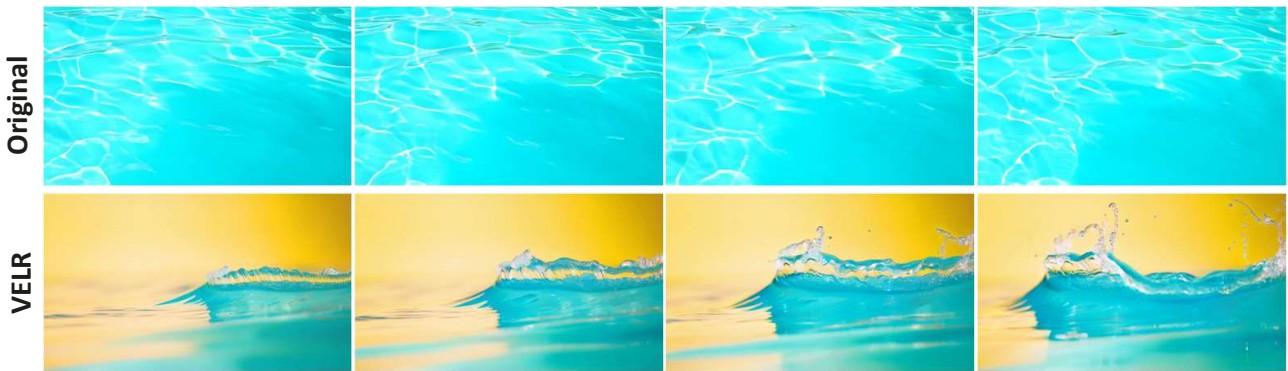

Splash of turquoise water in extreme slow motion, alpha channel included.

*Figure 13.* **Results of VELR on the validation set.** The concept of 'Splash of turquoise water' is more clearly expressed: compared to the nearly static pool in the original video, VELR produces a slow and realistic splash motion that **exhibits stronger temporal consistency**.

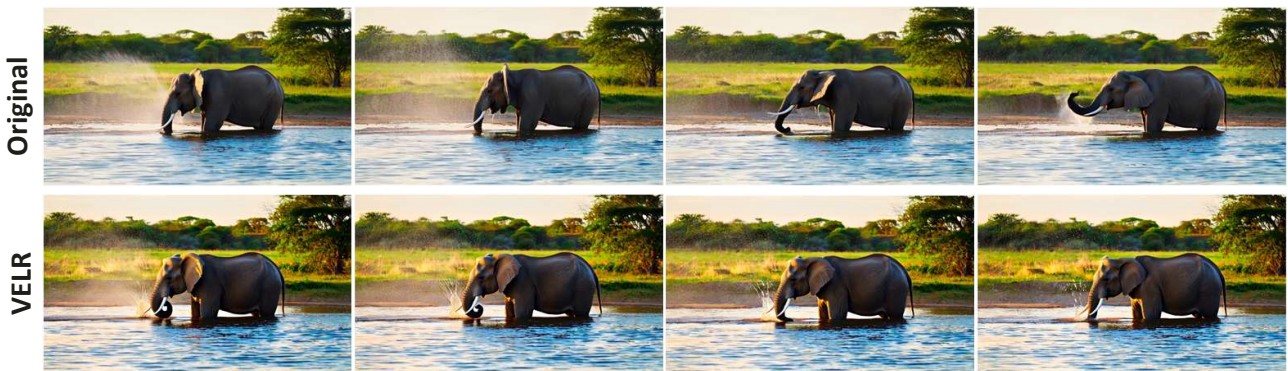

A big elephant spraying water with its trunk in the bright sunlight at a calm waterhole.

*Figure 14.* **Results of VELR on the validation set.** The elephant becomes more prominent, the background is richer and more realistic, and the concept of 'bright sunlight' is highlighted, with both the natural sunlight shining on the elephant and the background exhibiting warm tones of sunlight. The VELR video **illustrates higher visual quality and better aesthetics**.

## D. Test-time Scaling Experiment

We conducted a Test-Time Scaling experiment to demonstrate the inefficiency of previous ReFL update paradigms and the rationality and robustness of the truncated mid-step setting. Specifically, we added random noise to the latent representations at different denoising steps (step $= 0$ corresponds to pure noise, and step $= 50$ corresponds to the real video). This operation allows direct exploration of the generation range of videos at different steps, as adding noise is a more direct means than

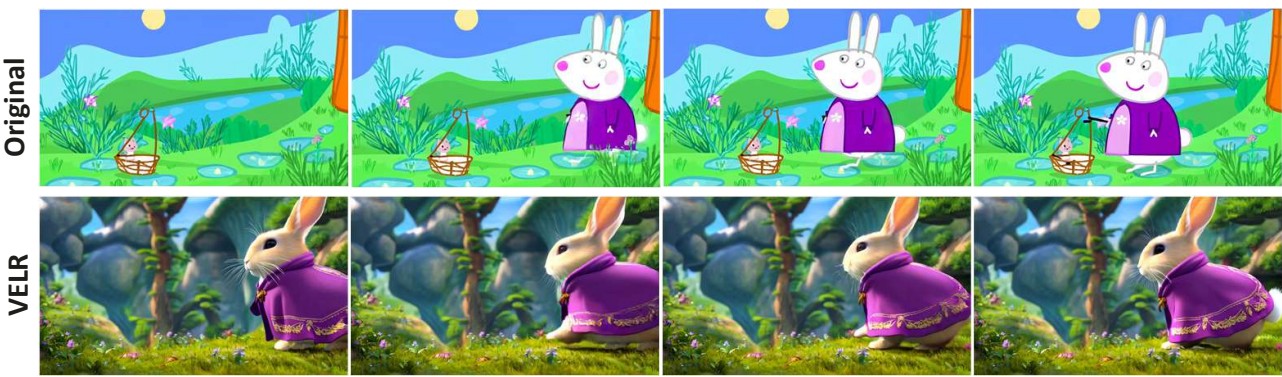

A fat rabbit wearing a purple robe walking through a fantasy landscape.

*Figure 15.* **Results of VELR on the validation set.** The overall video style shifts from anime to realistic, with the background more effectively reflecting the idea of a 'fantasy landscape'. The VELR video **showcases better semantic consistency**, as it avoids introducing irrelevant and anomalous objects (e.g., a basket not mentioned in the prompt) that appear in the original video. Moreover, VELR exhibits noticeably smoother motion, whereas the original video suffers from abrupt frame-to-frame transitions, indicating VELR's **superior temporal consistency**.

A fat rabbit wearing a purple robe walking through a fantasy landscape.

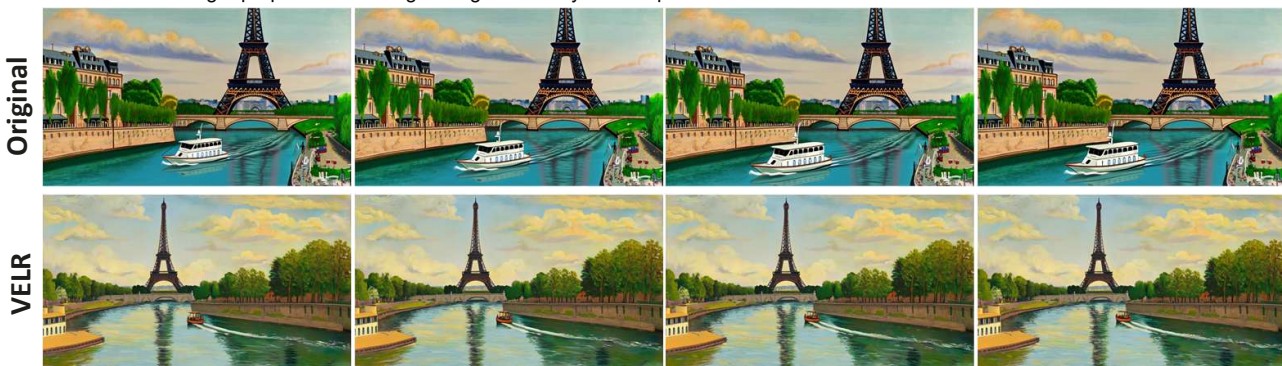

A boat sailing leisurely along the Seine River with the Eiffel Tower in background by Vincent van Gogh

*Figure 16.* **Results of VELR on the validation set.** The video style more closely matches the 'Vincent van Gogh' aesthetic specified in the prompt, and the boat maintains a physically plausible forward motion throughout the video. In contrast, the original video exhibits an anomalous backward movement in the final second, highlighting VELR's superior **temporal consistency**.

fine-tuning. Following the setting of DanceGRPO, we applied noise according to the DDPM noise scheduler. The results of adding noise at different steps are shown in the fig. 17.

The results show that early steps significantly affect the entire video, while mid-stage noise mainly influences the subject and fine details. After step 20, the video remains largely unchanged.

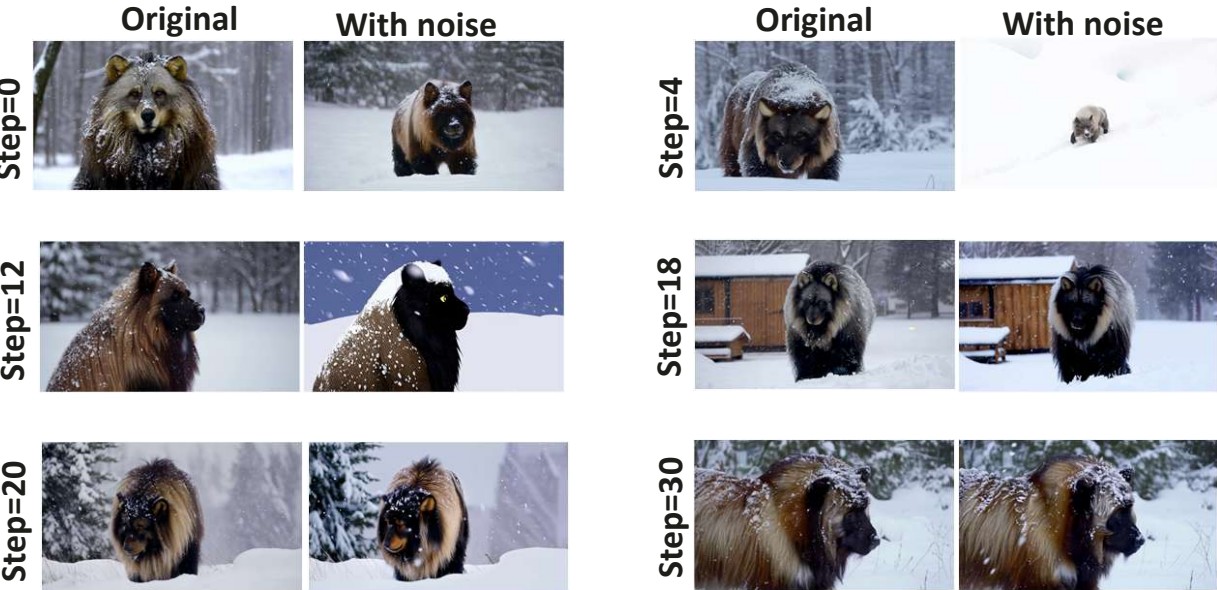

*Figure 17.* **Results of test-time-scaling.** Each row shows two steps: the left video is the original frame, and the right video is the noised frame.

