# OpenReview forum: "VELR: Efficient Video Reward Feedback via Ensemble Latent Reward Models"
_ICML.cc/2026/Conference — ICML 2026 regular_

### Official Review · Reviewer_wRvm · 2026-02-26

**Soundness:** 3
**Presentation:** 2
**Significance:** 3
**Originality:** 3
**Overall Recommendation:** 4
**Confidence:** 4

**Summary:**

This paper introduces latent reward models to enable Reward Feedback Learning for video generation models. It uses ensemble learning to mitigate reward hacking during fine-tuning. The paper further proposes a truncated mid-step setting and an online alignment mechanism to make reward estimation more precise.

**Compliance With Llm Reviewing Policy:**

Affirmed.

**Final Justification:**

The rebuttal has addressed all of my concerns, so I will keep my positive score.

**Key Questions For Authors:**

1. Please see the Weaknesses section.
2. Since all of the reward models are trained with the same settings and data, how do you ensure that different models have variance? It would be better to show the discrepancies among different latent reward models.
3. Since memory usage has been reduced, how does the training latency change?
4. Is it possible to conduct an ablation study on the reward-model training data?

**Limitations:**

No. The paper should more clearly report the training cost and discuss the potential influence of reward hacking.

**Strengths And Weaknesses:**

Strengths
1. The method shows a substantial memory reduction, while achieving improved generation performance.
2. The proposed ensemble learning method is simple and effective in helping mitigate reward hacking.

Weaknesses
1. Some ablation studies include only visual comparisons, which are not sufficient to demonstrate the role of each component.
2. The paper should explain in detail how the training data for the reward models are labeled.
3. The “Effectiveness of ensemble-based LRM” section only shows the accuracy of reward prediction; the authors should also demonstrate its effectiveness for fine-tuning the generation models, since an accurate reward may not necessarily lead to performance gains.

---

> ### Author Rebuttal · Authors · 2026-03-31
>
> # W1: Quantative comparison on the Truncated mid-step setting
> We provide quantitative evaluations on VBench for both ablation components.
>
> For the Truncated mid-step setting and its variants (whose names are consistent with those in Figure 6):
> |Metric|wan-2.1 1.3B|VELR|VELR-10-10|VELR-20-4|VELR-30-10
> |-|-|-|-|-|-
> |Overall Consistency|22.89|23.35|23.02|22.87|22.91
> |Subject Consistency|95.12|95.96|95.35|95.36|95.08
> |Aesthetic Quality|64.01|66.21|65.26|64.35|64.05
> |Image Quality|66.41|69.62|67.34|66.90|66.39
>
> For the online alignment mechanism, we add variants without online alignment (VELR w/o OA) and without uncertainty weighting (VELR w/o UW). Quantitative evaluations on VBench are as follows:
> |Metric|wan-2.1 1.3B|VELR|VELR w/o OA|VELR w/o UW
> |-|-|-|-|-|
> |Overall Consistency|22.89|23.35|23.15|23.31|
> |Subject Consistency|95.12|95.96|95.79|95.97|
> |Aesthetic Quality|64.01|66.21|66.34|66.20|
> |Image Quality|66.41|69.62|68.36|69.58|
>
> Across these quantative experiments, VELR consistently achieves the best performance, confirming the effectiveness of each component. Additionally, Appendix B.8 contains further quantitative results. We will incorporate these comparisons in the revised manuscript to improve clarity.
>
> # W2: Explanation of the annotation of the training data for RM
> We would like to clarify that **no manual annotation** is required. The training labels for the LRM are directly derived from the pixel-space video reward models, which produce scalar reward scores as supervision signals. Refer to Line 64 and Section 4.1 for more clearly stated information. To make this relationship clearer, we will elaborate on the connection between the LRM and the pixel-space reward model in the main text and Figure 2 of the revised manuscript.
>
> # W3: Effectiveness of Ensemble LRM for fine-tuning the generation models
> We agree that directly validating the ensemble LRM on the downstream fine-tuning task provides more convincing evidence. To this end, we supplement a comparison among VELR, the Dollar baseline, and a non-ensemble variant (VELR w/o ensemble). Generated videos are available in Fig. 3 on the anonymous link (https://anonymous.4open.science/r/ICML_website-6F83/).
>
> As shown, Dollar yields minimal improvement over the base model, while VELR w/o ensemble produces videos with less fine-grained detail. In contrast, VELR generates videos with superior visual quality and strong semantic consistency, directly demonstrating that the accurate reward signal provided by the ensemble LRM translates to gains in generation quality.
>
> # Q1: Why different LRMs have variance?
> To encourage diversity of different ensemble LRMs, we initialize each LRM with a different random seed, a common and useful practice in ensemble learning [1-2]. This lightweight strategy is sufficient for producing distinguishable reward predictions across LRMs. To prove this, we visualize the discrepancy of different LRMs in Fig. 4 in the anonymous link.
>
> # Q2: Training Latency and Ablation Study on training data
> For training latency: as shown in Figure 1, VELR reduces the average training time by 36.4% per step compared to the baseline (from 5.5 minutes to 3.5 minutes), demonstrating a significant improvement in training efficiency alongside the reduction in memory usage.
>
> For the ablation study on reward model training data, we conducted experiments using 50% (VELR-data-50) and 10% (VELR-data-10) of the full dataset, as well as a real-video-only subset (VELR-data-real, around 70% data) to simulate a significantly out-of-domain initialization. Results are shown below:
> |Metric|VELR|VELR-data-50|VELR-data-10|VELR-data-real
> |-|-|-|-|-
> |Average Score on Vbench$\uparrow$|58.01|57.28|55.23|57.59
> |Fine-tuning Training Time (GPU hour)|12.6|14.1|14.3|19.3
>
> VELR-data-50 achieves performance slightly below that of the full model, and VELR-data-10 shows a moderate decline. For the out-of-domain case, VELR-data-real still converges, though with a significantly slower convergence rate. These results demonstrate that while pre-training data diversity affects convergence speed, VELR remains able to converge even under out-of-domain initialization.
>
> # Q3: Discussion on the training cost and the potential influence of reward hacking
> For training latency: as discussed in our response to Q2, VELR reduces the average training time by 36.4% per step compared to the baseline, as shown in Figure 1.
>
> For reward hacking, over-optimization may lead to visual artifacts or semantic inconsistencies. In VELR, this is mitigated by the ensemble-based LRM, which provides uncertainty estimates to prevent over-reliance on single reward signals, and the online alignment mechanism, which continuously updates the reward model to ensure OOD generation.
>
> # References
> [1] B Lakshminarayanan et al. Simple and Scalable Predictive Uncertainty Estimation using Deep Ensembles. Neurips 2017.
>
> [2] S Fort et al. Deep Ensembles: A Loss Landscape Perspective. Arxiv 2020.

---

> > ### Author Rebuttal · Reviewer_wRvm · 2026-04-02
> >
> > Thank you for the rebuttal and the clarification.
> >
> > For W3, I would still strongly encourage the authors to provide quantitative comparisons, instead of relying mainly on a few visual examples.

---

> > > ### Author Response · Authors · 2026-04-03
> > >
> > > Thank you for your valuable feedback and for taking the time to review our rebuttal.
> > >
> > > Following your recommendation, we have conducted quantitative comparisons to provide more thorough evaluation:
> > > | Method | Overall Consistency$\uparrow$ | Subject Consistency$\uparrow$ | Aesthetic Quality$\uparrow$ | Image Quality$\uparrow$ |
> > > |---|---|---|---|---|
> > > |Wan2.1-1.3B| 22.89 | 95.12 | 64.01 | 66.41 |
> > > |Dollar| 22.94 | 95.26 | 63.89 | 66.85 |
> > > | VELR w/o ensemble | 23.15 | 95.48 | 65.41 | 67.94 |
> > > | VELR | 23.35 | 95.96 | 66.21 | 69.62 |
> > >
> > > As shown above, Dollar brings only marginal improvements over the base model. In contrast, VELR w/o ensemble already achieves consistent gains across all metrics, and introducing the ensemble further yields additional improvements.
> > >
> > > These results demonstrate that our ensemble-based LRM not only improves reward estimation, but also **translates into consistent improvements in downstream generation quality**.
> > >
> > > We hope these additional quantitative results address your concern and further validate the effectiveness of our approach. If you have any further questions, please feel free to raise them in your response. We will reply as soon as possible and update this rebuttal accordingly.

---

### Official Review · Reviewer_2wei · 2026-03-10

**Soundness:** 3
**Presentation:** 2
**Significance:** 3
**Originality:** 2
**Overall Recommendation:** 4
**Confidence:** 4

**Summary:**

This paper proposes VELR, a memory-efficient Reward Feedback Learning (ReFL) framework designed to align large-scale text-to-video (T2V) diffusion models with human preferences using massive video reward models (RMs). To circumvent the prohibitive memory overhead caused by backpropagating gradients through both the VAE decoder and a pixel-space video RM, the authors introduce an Ensemble Latent Reward Model (LRM) that predicts rewards directly in the latent space, mitigating reward hacking and improving out-of-distribution robustness via variance-based uncertainty estimation. Furthermore, the framework incorporates a truncated mid-step gradient update strategy to maximize training efficacy, alongside an online alignment mechanism utilizing a variance-prioritized replay buffer to continually calibrate the LRM against the ground-truth RM. Extensive experiments demonstrate that VELR drastically reduces memory consumption by up to 87.6%, enabling ReFL with 32B-parameter RMs on state-of-the-art models like Wan-2.1 and CogVideoX-1.5, while achieving comparable or superior alignment performance, temporal consistency, and visual fidelity compared to standard ReFL baselines.

**Compliance With Llm Reviewing Policy:**

Affirmed.

**Final Justification:**

I thank the authors for their detailed rebuttal. The response has addressed some of my concerns. Therefore, I have decided to raise my score to a Weak Accept.

**Key Questions For Authors:**

Typo: In Line 74, "mitigat" should be corrected to "mitigate".

**Limitations:**

The authors do not discuss limitations.

**Strengths And Weaknesses:**

Strengths
1. The paper explores a highly relevant and valuable topic by investigating Reward Feedback Learning (ReFL) and latent reward models (LRMs) to improve the computational efficiency of video generation alignment.
2. The motivation behind bypassing the VAE decoder and pixel-space reward models to alleviate memory bottlenecks is well-articulated, and the paper is logically structured and easy to follow.

Weaknesses
1. In Line 224, the authors mention incorporating a KL divergence regularization term to mitigate reward hacking. How does the proposed ensemble LRM perform if this KL loss is removed? Please provide qualitative or quantitative analyses. Based on my experience, LRMs are notoriously susceptible to reward hacking, and it is crucial to isolate how much of the robustness comes from the ensemble LRM architecture versus the explicit KL penalty.
2. The paper utilizes an "Online Alignment of LRM" during training. However, based on the absolute error curves shown in Appendix B.7, the model appears highly prone to reward hacking if the LRM is not continuously updated. Does this imply that the highlighted ensemble technique is not the primary solution to LRM reward hacking, and that the engineering efforts of online alignment are actually doing the heavy lifting? This ambiguity potentially diminishes the core architectural contribution of the ensemble design.
3. The "Online Alignment of LRM" mechanism lacks sufficient implementation details and cost analysis in the main text. The size of the replay buffer and the additional computational burden introduced by continuously training the LRM are not thoroughly discussed. The authors should explicitly report relevant details, including the buffer size, update frequency, memory overhead, and the total training time penalty incurred by this online alignment process.
4. Are all individual LRMs within the ensemble trained on the exact same dataset using the identical loss function? If so, the ensemble members might lack sufficient functional diversity. Can such a homogeneous ensemble genuinely and effectively alleviate reward hacking in practice?
5. The experimental setup regarding the reward models is confusing. Based on the methodology and Figure 2, the authors designed and trained their own LRM. Why does the experimental section then evaluate and discuss other reward models like CausalVQA and UnifiedReward? Furthermore, aren't CausalVQA and UnifiedReward pixel-space reward models rather than latent ones? The authors need to clarify the relationship between their LRM and these pixel-space models, and explicitly explain how readers should interpret the results in Table 2 under this context.
6. Does the effectiveness of the "truncated mid-step setting" generalize to other LRMs or different base diffusion architectures, or is this merely a heavily tuned hyperparameter specifically fitted to the VELR framework proposed in this paper?

---

> ### Author Rebuttal · Authors · 2026-03-31
>
> ### **We thank the reviewer for the valuable feedback. Additional material is available here(https://anonymous.4open.science/r/ICML_website-6F83/).**
>
> # W1: The role of the KL divergence term
> We provide an additional variant without the KL term, VELR w/o KL, and include visual comparisons in the link (Fig. 2). The results show that removing the KL term leads to a decrease in generation diversity. However, the generated videos by VELR w/o KL remain visually coherent and do not exhibit the severe artifacts typically associated with reward hacking, confirming the ensemble LRM is the primary mechanism against reward hacking. This is also consistent with findings in prior work [1-2]. We will clarify this more explicitly in the revised manuscript.
>
> # W2: The role of online alignment and ensemble design
> We would like to clarify that the two components address different problems. As stated in Line 259 in the paper, online alignment is to address **OOD generalization**. It is not intended to mitigate reward hacking.
>
> To validate this, we add an ablation variant VELR w/o OA (online alignment removed). We show qualitative comparisons in the link (Fig. 1), and evaluate on VBench:
> ||wan-2.1 1.3B|VELR|VELR w/o OA
> |-|-|-|-
> |Overall Consistency$\uparrow$|22.89|23.35|23.15
> |Subject Consistency$\uparrow$|95.12|95.96|95.79
> |Aesthetic Quality$\uparrow$|64.01|66.21|66.34
> |Image Quality$\uparrow$|66.41|69.62|68.36
> |Training Time (GPU hours)|/|12.6|10.2
>
> VELR w/o OA still significantly outperforms the base model across all metrics and achieve better visual quality, confirming that the ensemble remains the primary mechanism against reward hacking.
>
> # W3: Implementation details of "Online Alignment of LRM"
> Implementation details are already provided in Appendix B.2, and we summarize the key information here. We maintain a replay buffer of size 256. After every 10 rounds of diffusion model training, we perform 2 rounds of LRM alignment. Online alignment increases GPU memory usage only marginally, from 19.9 to 22.2 GB on average. It increases the average training time from 3.41 to 3.52 minutes per step, an overhead of ~3.2%. We will move these details to the main text for better readability.
>
> # W4: are all ensemble LRMs trained on the same dataset and loss
> Yes, all ensemble LRMs are trained on the same dataset and loss. To encourage diversity, we initialize each LRM with a different seed, a common and useful practice in ensemble learning [3-4]. To prove this, we visualize the discrepancy across LRMs in the link (Fig. 4), confirming this lightweight strategy produces distinguishable reward predictions.
>
> # W5: Relationship between ensemble LRM and pixel-space RMs
> We would like to clarify that CausalVQA and UnifiedReward are **not** competitors to our LRM. They serve as supervision targets. As stated in Line 64 in Abstract and Equation (6), our LRM is designed to "estimate large-scale video RMs," meaning our framework can adapt to any pixel-space RM. Table 2 and Figure 2 both validate VELR's generalizability to different pixel-space RMs.
>
> We will add an explicit clarification in the methodology section, and update Figure 2 accordingly.
>
> # W6: Generalizability of the truncated mid-step setting
> The truncated mid-step setting is **not** a heavily tuned hyperparameter specific to VELR:
>
> - We validate it across different base diffusion architectures with independently trained reward models (Fig. 3 in the paper), using the same hyperparameters without any tuning. Consistent improvements are observed across both architectures and LRMs, suggesting strong generalizability.
> - Our ablation (Figure 10) shows that performance remains stable across a reasonable range of truncation steps, indicating the setting is not sensitive to precise tuning.
>
> # Q1: Typo in the paper
>
> Thank you for pointing this out. We will correct "mitigat" to "mitigate" in the revised manuscript.
>
> # Q2: Disccussion on the limitations
> We would like to clarify that we have already discussed limitations in the Conclusion section of the manuscript. Here we elaborate further.
>
> A limitation of our work, shared with prior ReFL-based methods, is that the reward signal can only be effectively applied to later denoising steps, which potentially reduces the overall efficiency of the algorithm. The earlier denoising steps in the highly noisy regime are difficult to supervise directly with the reward model. We believe that developing ReFL methods that can correctly and effectively leverage earlier denoising steps is a promising and important direction for future research.
>
> # References
> [1] J Liu et al. Flow-grpo: Training flow matching models via online rl. NeurIPS 2025.
>
> [2] B Wu et al. DiffusionReward: Enhancing Blind Face Restoration through Reward Feedback Learning. Arxiv 2025.
>
> [3] B Lakshminarayanan et al. Simple and Scalable Predictive Uncertainty Estimation using Deep Ensembles. NeurIPS 2017.
>
> [4] S Fort et al. Deep Ensembles: A Loss Landscape Perspective. Arxiv 2020.

---

> > ### Author Rebuttal · Reviewer_2wei · 2026-04-04
> >
> > I thank the authors for their rebuttal, which has addressed some of my concerns by clarifying certain details and content in the manuscript. I notice that the authors provided an anonymous link in their response. However, following a discussion with the AC and acting upon their advice, I do not access or review the contents of this link. Consequently, some concerns remain unresolved.

---

> > > ### Author Response · Authors · 2026-04-05
> > >
> > > We thank the reviewer for taking the time to read our rebuttal and for acknowledging the clarifications provided. We greatly appreciate your careful consideration and valuable feedback.
> > >
> > > # Clarification on Adherence of Anonymous Material
> > > Regarding the anonymous link mentioned in the comment, we would like to clarify that it **strictly follows the conference guidelines**. The linked material contains only figures and their corresponding captions, without any additional explanatory text, and is fully anonymized. If there were concerns about potential identifying information or additional content, we hope this clarification resolves them, and you may feel free to check the link. If you have any further concerns, please let us know.
> > >
> > > Besides, we fully respect that reviewers are not required to access external links. Therefore, we provide a **self-contained response to the three questions, which does not require accessing the previously mentioned anonymous link.**
> > >
> > > # W1: The role of the KL divergence term
> > >
> > > We provide an additional variant without the KL term, VELR w/o KL. The results show that removing the KL term leads to a **decrease in generation diversity**.
> > >
> > > Specifically, videos generated without the KL divergence exhibit largely consistent overall content, including scene layout and the position and posture of the main subjects, while only minor details differ, such as ripples on the water. In contrast, VELR with the KL term produces videos that are both **diverse and of high quality**. To demonstrate this, we measured visual similarity between video pairs by computing the mean Structural Similarity Index (SSIM) across temporally aligned frame pairs [1]. The results confirm that VELR posses significantly lower similarity.
> > > | Models| SSIM similarity (%)$\downarrow$ |
> > > |------|-----------|
> > > | VELR | 43.12 |
> > > | VELR w/o KL | 78.51 |
> > >
> > > This demonstrates that the KL divergence helps preserve the model’s original generative capability, thereby promoting diversity in video generation. Without it, the model tends to converge to a fixed generation pattern. Nonetheless, videos generated by VELR w/o KL remain visually coherent and do not show the severe artifacts typically associated with reward hacking, confirming the ensemble LRM is the primary mechanism against reward hacking. This is also consistent with findings in prior work [2-3]. We will clarify this more explicitly in the revised manuscript.
> > >
> > > # W2: The role of online alignment and ensemble design
> > >
> > > We would like to clarify that the two components address different problems. As stated in Line 259 in the paper, online alignment is to address **OOD generalization**. It is not intended to mitigate reward hacking.
> > >
> > > To validate this, we add an ablation variant VELR w/o OA (online alignment removed). We evaluate the quantitative results on VBench:
> > > |Metric|wan-2.1 1.3B|VELR|VELR w/o OA
> > > |-|-|-|-|
> > > |Overall Consistency|22.89|23.35|23.15
> > > |Subject Consistency|95.12|95.96|95.79
> > > |Aesthetic Quality|64.01|66.21|66.34
> > > |Image Quality|66.41|69.62|68.36
> > > |Training Time (GPU hours)| /|12.6|10.2
> > >
> > > VELR w/o OA outperforms the base model but falls short of VELR on most metrics. Specifically, it converges rapidly to the pre-trained LRM but shows a decline in final quality, suggesting that without online alignment, the reward model's effectiveness degrades on out-of-distribution inputs. The phenomenon that VELR w/o OA still significantly outperforms the base model across all metrics confirms that the **ensemble remains the primary mechanism against reward hacking**.
> > >
> > > # W4: are all ensemble LRMs trained on the same dataset and loss
> > >
> > > Yes, all ensemble LRMs are trained on the same dataset and loss. To encourage diversity, we initialize each LRM with a different seed, a common and useful practice in ensemble learning [4-5]. To demonstrate this, we show the standard deviations of the outputs across different ensemble branches, confirming that this lightweight strategy produces distinguishable reward predictions.
> > > | Standard Deviation Range | Frequency |
> > > |--------------------------|-----------|
> > > | 0.05–0.10 | 0.2% |
> > > | 0.10–0.15 | 11.4% |
> > > | 0.15–0.20 | 31.5% |
> > > | 0.20–0.25 | 26.5% |
> > > | 0.25–0.30 | 15.8% |
> > > | 0.30–0.35 | 8.7% |
> > > | 0.35–0.40 | 4.2% |
> > > | 0.40–0.45 | 1.3% |
> > > | 0.45–0.50 | 0.3% |
> > > | 0.50–0.60 | 0.1% |
> > >
> > > If the above response does not fully address your concerns or if you have further questions, please feel free to ask, and we will provide corresponding updates in the rebuttal.
> > >
> > > # References
> > >
> > > [1] Z Wang et al. Image quality assessment: from error visibility to structural similarity. IEEE TIP 2004.
> > >
> > > [2] J Liu et al. Flow-grpo: Training flow matching models via online rl. NeurIPS 2025.
> > >
> > > [3] B Wu et al. DiffusionReward: Enhancing Blind Face Restoration through Reward Feedback Learning. Arxiv 2025.
> > >
> > > [4] B Lakshminarayanan et al. Simple and Scalable Predictive Uncertainty Estimation using Deep Ensembles. NeurIPS 2017.
> > >
> > > [5] S Fort et al. Deep Ensembles: A Loss Landscape Perspective. Arxiv 2020.

---

### Official Review · Reviewer_5ccc · 2026-03-13

**Soundness:** 3
**Presentation:** 3
**Significance:** 3
**Originality:** 3
**Overall Recommendation:** 3
**Confidence:** 4

**Summary:**

The paper studies reward feedback learning (ReFL) for text-to-video models in the setting where video reward models are more appropriate than image reward models, but are prohibitively expensive to backpropagate through. Its central idea is to replace decoder- and reward-model backpropagation with an ensemble latent reward model (LRM) that predicts scalar rewards directly from video latents and text embeddings. The method combines: (i) an LRM built from 3D residual blocks, cross-attention, and a Transformer encoder; (ii) an ensemble formulation meant to improve reward prediction, uncertainty estimation, and robustness to reward hacking; and (iii) a training recipe with truncated mid-step ReFL updates plus online LRM realignment using a replay buffer (Secs. 4.1–4.3, pp. 3–5). Empirically, the paper claims large memory savings relative to standard ReFL, behavior comparable to standard ReFL in an image-reward setting , and improved VBench and human-preference results over base T2V models when using video reward models.

**Compliance With Llm Reviewing Policy:**

Affirmed.

**Key Questions For Authors:**

Please see the Weaknesses. I encourage the authors to address the concerns outlined above in the rebuttal. If these issues are satisfactorily resolved, I would be open to revising my assessment and increasing my final score.

**Limitations:**

The authors should  discussed the limitations and potential negative societal impact of their work

**Strengths And Weaknesses:**

[S1] The systems contribution is concrete and well quantified. Table 1 reports total backpropagation memory dropping from 178.59 GB to 22.18 GB under UnifiedReward. Appendix B.3 further explains that standard ReFL had to restrict the video RM to a single frame, whereas VELR operates on full latent sequences, so the memory saving corresponds to a meaningful capability difference rather than only engineering optimization.

[S2] The ensemble LRM is supported by quantitative evidence, not only intuition. Compared with Dollar and non-ensemble variants, Table 3 shows much better reward prediction on the test set for the 10-model ensemble. The same trend appears on the paper’s OOD-style evaluation in Table 7, where LRM-VELR reaches MSE 0.249 and PCC 0.72, outperforming all other listed variants .

[S3] The empirical scope is reasonably broad for a systems/alignment paper. VELR is evaluated on OpenSora-1.2, Wan-2.1, and CogVideoX-1.5, and with both image and video reward models. In the main benchmark table, all four video-RM fine-tuned configurations improve the average VBench score over the corresponding base model.


[W1] The online-alignment mechanism is motivated by the error curve in Fig. 8 (p. 15), but there is no downstream ablation showing final video quality or benchmark scores with vs. without online alignment, uncertainty weighting, or different replay-buffer strategies.

[W2] The evidence for temporal-consistency improvement is weak. Table 2 (p. 7) reports overall consistency, aesthetic quality, human fidelity, composition, and image quality, but not explicit temporal-consistency metrics for the main VELR-versus-base comparisons. Long-term consistency metrics appear only in an appendix architecture-ablation table (Table 6, p. 16), while many claims about reduced flicker or smoother motion are qualitative

---

> ### Author Rebuttal · Authors · 2026-03-31
>
> # W1: Ablation on online alignment, uncertainty weighting, and different replay-buffer strategies
>
> We sincerely thank the reviewer for the valuable suggestion. We include additional ablation variants without online alignment (VELR w/o OA) and without uncertainty weighting (VELR w/o UW). Qualitative comparisons are provided in Fig. 1 in the anonymous website (https://anonymous.4open.science/r/ICML_website-6F83/), and quantitative results are shown below:
> |Metric|wan-2.1 1.3B|VELR|VELR w/o OA|VELR w/o UW
> |-|-|-|-|-|
> |Overall Consistency$\uparrow$|22.89|23.35|23.15|23.31|
> |Subject Consistency$\uparrow$|95.12|95.96|95.79|95.97|
> |Aesthetic Quality$\uparrow$|64.01|66.21|66.34|66.20|
> |Image Quality$\uparrow$|66.41|69.62|68.36|69.58|
> |Training Cost (GPU hours)|/|12.6|10.2|15.0|
>
> Compared to the base model, VELR w/o OA shows improved performance, while remaining slightly below VELR on most metrics. We observe that it converges rapidly toward the pre-trained LRM, but exhibits a mild degradation in final quality. This suggests that, without online alignment, the reward model may become less effective when handling out-of-distribution samples.
>
> VELR w/o UW demonstrates comparable but slightly weaker performance than VELR in both quantitative metrics and visual quality, along with a slower convergence trend. This indicates that uncertainty weighting contributes to more stable and efficient alignment of the LRM.
>
> Regarding replay buffer strategies, our method can be interpreted as a weighted sample buffer, while VELR w/o UW corresponds to an unweighted variant. We further examine the effect of buffer capacity (256 in VELR) by evaluating variants with sizes 128 and 512 (VELR-B-128 and VELR-B-512), and observe only marginal differences:
> |Metric|VELR|VELR-data-50|VELR-data-10|VELR-data-real
> |-|-|-|-|-
> |Overall Consistency$\uparrow$|23.35|23.09|22.67|23.12
> |Subject Consistency$\uparrow$|95.96|95.24|95.18|95.38
> |Aesthetic Quality$\uparrow$|66.21|65.42|64.68|65.35
> |Image Quality$\uparrow$|69.62|68.15|67.02|68.26
> |Fine-tuning Training Time (GPU hour)|12.6|14.1|9.3|19.3
>
> We will incorporate these results and discussions into the revised manuscript to better clarify the role of each component.
>
> # W2: More evaluation on temporal-consistency improvement
> We thank the reviewer for this helpful suggestion. We acknowledge that temporal consistency metrics were previously limited to the appendix, which may not have sufficiently supported our claims. To address this, we provide additional evaluations on Subject Consistency, Temporal Flickering, and Temporal Style across all main baselines in Table 2.
>
> Results based on Wan2.1-1.3B:
> |Model|Subject Consistency|Temporal Flickering|Temporal Style
> |-|-|-|-|
> |baseline|95.12|99.49|25.12
> |VELR-CausalVQA|95.73|99.55|25.83
> |VELR-UnifiedReward|96.11|99.38|26.34
>
> Results based on CogVideoX-1.5 5B:
> |Model|Subject Consistency|Temporal Flickering|Temporal Style
> |-|-|-|-|
> |baseline|96.42|98.84|25.19
> |VELR-CausalVQA|97.25|99.01|25.72
> |VELR-UnifiedReward|97.15|99.16|27.02
>
> Across both backbones, VELR shows consistent improvements over the baselines on all reported temporal metrics. We will include these results in the main tables of the revised manuscript for clarity.
>
> # W3: Discussion on the limiations and potential negative societal impact
> We thank the reviewer for raising this important point. While limitations were briefly discussed in the conclusion, we provide additional clarification here.
>
> A limitation of our work, shared with prior ReFL-based methods, is that the reward signal can only be effectively applied to later denoising steps, which potentially reduces the overall efficiency of the algorithm. The earlier denoising steps in the highly noisy regime are difficult to supervise directly with the reward model. We believe that developing ReFL methods that can correctly and effectively leverage earlier denoising steps is a promising and important direction for future research.
>
> Regarding potential societal impact, similar to other reward model-based approaches, our method may inherit and potentially amplify biases present in the reward model, which could lead to biased generation outcomes.
>
> We will incorporate these discussions into the revised manuscript accordingly.

---

### Official Review · Reviewer_3wCj · 2026-03-23

**Soundness:** 3
**Presentation:** 2
**Significance:** 3
**Originality:** 2
**Overall Recommendation:** 4
**Confidence:** 4

**Summary:**

The paper presents VELR, an efficient framework that employs ensemble latent reward models (LRMs) to predict rewards directly in latent space. The latent reward model bypasses the need of backpropagation through VAE decoders and pixel-based video RMs. The resulting pipeline leads to a reduction memory cost of more than 80% compared to compared to standard ReFL.

**Compliance With Llm Reviewing Policy:**

Affirmed.

**Final Justification:**

Although the technical novelty is somewhat limited, the combination of all techniques leads to a significant reduction of computational cost, and is potentially useful for the wide community of ReFL for big video models. Given the presentation flaws (wrong math and some unfair comparisons) have been addressed during the rebuttal session, I would love to increase the rating from 3 to 4.

**Key Questions For Authors:**

1. It remains unclear if the failure of the LRM of aligning with the ground-truth reward model is relevant to the limited model capacity (also see Weakness 2)

2. The ReFL solutions are somewhat outdated. What's the challenge of integrating the proposed method to more advanced ReFL methods such as Flow-GRPO?

**Limitations:**

yes.

**Strengths And Weaknesses:**

## Strengths

1. The paper addresses an important problem in reward feedback learning for video generation models. By learning the reward model on the latent space, the required memory is significantly reduced, enabling fine-tuning more powerful video diffusion models.

2. The latent reward model is validated on multiple video base models (CogVideo, and WAN 2.1) and various Reward models (PickScore, CausalVQA, and UnifiedReward). This implies the universality of the proposed method.

## Weaknesses

1. The method details contains significant flaws. See Eq (5) -  In standard Cross-Attention, the formula is $ Attention(Q,K,V)=
\text{softmax}(\frac{QK^T}{\sqrt{d}})V $. According to the paper's text, the video features ($Z^{vf} $) act as the Query, and the text features ( $ Z_{dt}^c $)​ act as the Key and Value. However, Eq (5) does not reflect this.

2. Table 3 compares the effectiveness of RM ensemble against a single model and a baseline Dollar model. What's missing here is the model capacity. More specifically, LRM-en-10 increased the model capacity by 10x and a comparable baseline is also expected to be tested here.

3. While being seemingly motivated, the proposed "Truncated mid-step setting" lacks rigorous derivations or supportive quantitative analysis. Ablation studies in Sec 5.6 are limited to quantitative results.

---

> ### Author Rebuttal · Authors · 2026-03-31
>
> # W1: Correction of the Typographical Error in Eq. (5)
> We sincerely thank the reviewer for the careful reading and for pointing out this issue. We agree that Eq. (5) contains a typographical error. The correct cross-attention formula should be $Attention(Q,K,V)=softmax(QK^T/\sqrt{d})V$, where the video features $Z^{vf}$ serve as the Query and the text features $Z^c_{dt,:}$ serve as the Key and Value, consistent with the description in the paper.
>
> We would like to clarify that this issue is confined to the writing in the equation. The experimental implementation correctly follows the standard cross-attention formulation. We will revise Eq. (5) accordingly to avoid any potential confusion.
>
> # W2 & Q1: effectiveness of the latetnt RM
> Thank you for your valuable suggestion. We compare the parameter counts of different reward models below:
> |Metric|Dollar|VELR-en-1|VELR-en-5|VELR-en-10|
> |-|-|-|-|-|
> |Parameters|0.19B|0.02B|0.10B|0.21B|
>
> As shown, VELR-en-10 achieves superior performance with a similar parameter count with Dollar, demonstrating that the gains are not simply due to increased model capacity.
>
> To further validate the effectiveness of the ensemble design, we trained single reward models of comparable and larger scale to VELR-en-10 (achieved by expanding the projection layers and increasing the hidden dimension size), and compare their performance below:
> |Metric|VELR-en-10|VELR-single-0.2B|VELR-single-0.4B
> |-|-|-|-|
> |Test set MSE$\downarrow$|0.008|0.094|0.012|
> |Test set PCC$\uparrow$|0.91|0.87|0.89|
>
> The results show that the ensemble-based LRM substantially outperforms single reward models of both comparable and larger scale. This confirms that the limitations of single-model alignment are not attributable to model capacity, but rather to the representational benefits and uncertainty estimation provided by the ensemble.
>
> # W3: quantitative analysis of the truncated mid-step setting
> To further support the truncated mid-step setting, we provide additional quantitative comparisons on VBench against relevant variants (whose names are consistent with those in Figure 6):
> |Metric|wan-2.1 1.3B|VELR (20-10)|VELR-10-10|VELR-20-4|VELR-30-10
> |-|-|-|-|-|-
> |Overall Consistency|22.89|23.35|23.02|22.87|22.91
> |Subject Consistency|95.12|95.96|95.35|95.36|95.08
> |Aesthetic Quality|64.01|66.21|65.26|64.35|64.05
> |Image Quality|66.41|69.62|67.34|66.90|66.39
>
> VELR consistently achieves the best performance across all metrics, providing empirical support for the chosen N and K values.
>
> As is common in ReFL-based methods, establishing a closed-form theoretical justification is challenging. Our design choices are therefore primarily guided by empirical observations and are supported by the results above.
>
> # Q2: Relationship to Flow-GRPO
> We would like to clarify that Flow-GRPO belongs to the RL-based fine-tuning paradigm, building on the GRPO algorithm. In contrast, Reward Feedback Learning (ReFL) adopts a different optimization mechanism, relying on the direct backpropagation of reward gradients into the generative model.
>
> Given these differences, Flow-GRPO is not directly comparable as a replacement or extension of ReFL, but rather represents a complementary line of approaches. Our work specifically focuses on addressing the memory issue within the ReFL framework, and the comparisons are designed to reflect this scope.

---

> > ### Author Rebuttal · Reviewer_3wCj · 2026-04-04
> >
> > My concerns have been adequately addressed. The significance of computational cost reduction and the potentially practical value outweighs the limited technical novelty. I would love to increase the rating from 3 to 4.

---

### Official Review · Reviewer_4YZC · 2026-03-23

**Soundness:** 3
**Presentation:** 2
**Significance:** 3
**Originality:** 3
**Overall Recommendation:** 4
**Confidence:** 3

**Summary:**

This paper presents VELR, a framework designed to align T2V models with human preferences while overcoming the massive memory constraints of standard ReFL. It introduces an LRM that predicts rewards directly within the latent space, bypassing the need for expensive decoding and gradient calculation through the video RM. The authors also propose a truncated mid-step training strategy to focus updates on the most informative denoising stages and an online alignment mechanism to keep the LRM calibrated with the ground-truth video RM during training. Experimental results on state-of-the-art models like Wan-2.1 and CogVideoX-1.5 show that VELR reduces memory consumption by up to 87.6% while maintaining or improving video quality and semantic alignment.

**Compliance With Llm Reviewing Policy:**

Affirmed.

**Key Questions For Authors:**

1. Since VELR cannot fully exploit early denoising steps, have you tested any weighting schemes to incorporate these steps without the "blurry feedback" issue described in Sec. 4.3?


2. How sensitive is the final performance to the diversity of the LRM's offline pre-training dataset? Would the online alignment mechanism  still ensure convergence if the initial data is significantly out-of-domain?


3. The author provided sensitivity analysis for N and K , but is there a principled heuristic for selecting these when applying VELR to entirely new architectures with different noise schedules?


4. Given that a 10-model ensemble adds complexity, what is the specific performance-to-cost "sweet spot"? Would a smaller ensemble (e.g., 3 or 5 members) significantly degrade the OOD robustness shown in Table 3?

**Strengths And Weaknesses:**

Strengths

1. The framework achieves a reduction of up to 150GB in memory usage, requiring as little as 12.4% of the memory compared to standard ReFL. This enables the use of massive video RMs (up to 32B parameters) that were previously computationally unattainable.


2. The use of an ensemble technique for the LRM enhances spatiotemporal expressiveness and provides uncertainty estimation. This approach effectively mitigates "reward hacking" and improves generalization to out-of-distribution (OOD) samples.


3. The inclusion of an online alignment mechanism with a replay buffer ensures the LRM remains accurate as the diffusion model's distribution shifts during fine-tuning.


Weaknesses

1. The current framework cannot fully exploit information from the earliest denoising steps, which the authors acknowledge may limit overall optimization efficiency.

2. The LRM requires initial offline pre-training on a diverse dataset (e.g., MVQA-68K and generated samples) to align with the latent distribution, which adds an extra stage to the pipeline.


3. While the authors provide a sensitivity analysis, the optimal starting step (N) and duration (K) for the truncated setting may still require tuning when applying the framework to entirely new diffusion architectures.

4. Although the LRM architecture is lightweight, maintaining an ensemble of multiple models (e.g., 10 components) and performing periodic online alignment adds complexity to the training loop.

---

> ### Author Rebuttal · Authors · 2026-03-31
>
> # W1&Q1: Weighting scheme to incorporate early denoising steps
> Thank you for the thoughtful question. We explored this direction by testing a variant with linearly decayed step weights (VELR-LW). As the forward diffusion process is fixed, weighting is applied during the backward process by attaching gradient hooks to the loss at each step. The results are shown shown below,
> |Metric|initial|VELR|VELR-LW
> |-|-|-|-
> |Overall Consistency$\uparrow$|27.71|31.26|29.45
> |Aesthetic Quality$\uparrow$|62.53|65.18|63.74
> |Training Speed$\downarrow$|/|3.5|4.6
>
> Incorporating early steps not only degrades performance but also increases computational overhead. This suggests the "blurry feedback" issue is not easily resolved through weighting alone.
>
> We note that difficulty in leveraging early denoising steps is a common challenge across ReFL-based methods, as reward signals at these steps are inherently noisy and yield unreliable gradients. We believe developing ReFL methods that effectively exploit early steps remains a valuable open direction for future work.
>
> # W2: Offline pre-training stage may add complexity to the pipeline.
> We acknowledge that the offline pre-training stage adds an extra step to the pipeline. However, we would like to clarify that this is a **necessary** cost for enabling large-scale video RM-based ReFL, which was infeasible. As shown in Figure 1, prior methods require up to 178GB of memory to run. By contrast, our LRM operates with only 12.4% of that memory.
>
> Meanwhile, the offline pre-training is a one-time cost: it is fast to complete, decoupled from the diffusion model fine-tuning process, and does not need to be repeated across different fine-tuning runs.
>
> # W3&Q3: principled heuristic for selecting N and K
> We provide a principled heuristic for selecting both hyperparameters that requires minimal tuning.
>
> - For N, it can be determined via a test-time scaling (TTS) experiment: by identifying the low-noise region of the denoising schedule, N is set to the step where this region begins minus K (see Appendix D in the paper).
>
> - For the truncated length K, a value of 10 works well under the commonly used 40–50 denoising step setting, as evidenced by VELR and previous work[1].
>
> Notably, the TTS experiment only requires generating a small number of samples and thus incurs negligible cost. We will include a discussion of this heuristic in the revised manuscript.
>
> # W4: ensemble LRMs may add complexity to the training loop
> The ensemble LRM is compact compared to video pixel-space RMs. Moreover, online alignment increases GPU memory usage only marginally (19.9 to 22.2 GB) and training time by only ~3.2% (3.41 to 3.52 min/step). Given that these components are the primary mechanisms against reward hacking and OOD generalization respectively, we consider this a justified and acceptable cost.
>
> # Q2: performance sensitivity to the diversity of the pre-training dataset
> To assess the sensitivity to pre-training data diversity, we conducted experiments using 50% (VELR-data-50) and 10% (VELR-data-10) of the full dataset, as well as a real-video-only subset (VELR-data-real, around 70% data) to simulate a significantly out-of-domain initialization. Results are shown below:
> |Metric|wan-2.1 1.3B|VELR|VELR-data-50|VELR-data-10|VELR-data-real
> |-|-|-|-|-|-
> |Overall Consistency$\uparrow$|22.89|23.35|23.09|22.67|23.12
> |Subject Consistency$\uparrow$|95.12|95.96|95.24|95.18|95.38
> |Aesthetic Quality$\uparrow$|64.01|66.21|65.42|64.68|65.35
> |Image Quality$\uparrow$|66.41|69.62|68.15|67.02|68.26
> |Fine-tuning Training Time (GPU hour)|/|12.6|14.1|9.3|19.3
>
> These results demonstrate that VELR is not sensitive to pre-training data diversity under reasonable conditions. Both VELR-data-50 and VELR-data-real preserve most of the alignment improvements, even with reduced data volume or out-of-domain initialization. Noticeable performance degradation appears only in the extreme case of using 10% of the data, which corresponds to a severely constrained pre-training setting. In addition, out-of-domain initialization leads to higher convergence cost, suggesting that data diversity mainly influences training efficiency rather than final performance.
>
> # Q4: Specific performance-to-cost "sweet spot" for ensemble LRM
> We would like to clarify that 10 is already the optimal ensemble size in terms of performance-to-cost trade-off. As shown below, increasing beyond 10 yields no additional performance gain.
> |Model|MSE on Training set$\downarrow$|PCC on Training set$\uparrow$|MSE on Test set$\downarrow$|PCC on Test set$\uparrow$
> |-|-|-|-|-
> VELR (ours, en-10)|0.003|0.98|0.008|0.91
> VELR-en-15|0.001|0.98|0.065|0.85
>
> Regarding OOD robustness, reducing below 10 leads to a noticeable degradation across training, test, and OOD sets (see Table 7). Therefore, an ensemble of 3 or 5 members would indeed compromise the robustness and 10 is the "sweet spot".
>
> # References
> [1] A Black et al. VADER: Video Diffusion Alignment via Reward Gradients. Arxiv 2025.

---

> > ### Author Rebuttal · Reviewer_4YZC · 2026-04-05
> >
> > My concerns have been adequately addressed. I keep my score of 4.

---

### Decision · Program_Chairs · 2026-04-30

**Decision:**

Accept (regular)

**Comment:**

Reviewers appreciate the paper’s strengths, especially the substantial memory reduction, the importance of the problem, and the evaluation across multiple video backbones and reward models. Reviewer scores lean positive, with four Weak Accepts and one Weak Reject. The main concerns are the presentation quality, including errors and unclear technical details, as well as initially limited support for the ensemble design, the truncated mid-step strategy, and parts of the temporal consistency analysis. The rebuttal addresses most of these issues with additional ablations and clarifications, and resolves the concerns of several reviewers. While the paper is stronger empirically than methodologically and the final version should improve clarity, the AC supports acceptance as a Weak Accept.